# On the Inductive Biases of Demographic Parity-based Fair Learning Algorithms

**Haoyu Lei**[1]  **Amin Gohari**[2]  **Farzan Farnia**[1]

[1]Department of Computer Science and Engineering, The Chinese University of Hong Kong
[2]Department of Information Engineering, The Chinese University of Hong Kong

## Abstract

Fair supervised learning algorithms assigning labels with little dependence on a sensitive attribute have attracted great attention in the machine learning community. While the demographic parity (DP) notion has been frequently used to measure a model's fairness in training fair classifiers, several studies in the literature suggest potential impacts of enforcing DP in fair learning algorithms. In this work, we analytically study the effect of standard DP-based regularization methods on the conditional distribution of the predicted label given the sensitive attribute. Our analysis shows that an imbalanced training dataset with a non-uniform distribution of the sensitive attribute could lead to a classification rule biased toward the sensitive attribute outcome holding the majority of training data. To control such inductive biases in DP-based fair learning, we propose a sensitive attribute-based distributionally robust optimization (SA-DRO) method improving robustness against the marginal distribution of the sensitive attribute. Finally, we present several numerical results on the application of DP-based learning methods to standard centralized and distributed learning problems. The empirical findings support our theoretical results on the inductive biases in DP-based fair learning algorithms and the debiasing effects of the proposed SA-DRO method. The project code is available at `github.com/lh218/Fairness-IB.git`.

## 1 INTRODUCTION

A responsible deployment of modern machine learning frameworks in high-stake decision-making tasks requires mechanisms for controlling the dependence of their output on sensitive attributes such as gender and ethnicity. A supervised learning framework with no control on the dependence of the prediction on the input features could lead to discriminatory decisions that significantly correlate with the sensitive attributes. Due to the critical importance of the fairness factor in several machine learning applications, the study and development of fair statistical learning algorithms have received great attention in the literature.

A widely-used approach to fair supervised learning is to include a fairness regularization penalty term in the learning objective that quantifies the level of fairness violation according to a fairness notion. A standard fairness notion is the *demographic parity (DP)* aiming toward a statistically independent prediction variable $\widehat{Y}$ of a sensitive attribute $S$. Therefore, a DP-based fairness regularization metric should be a measure of the dependence of the prediction $\widehat{Y}$ on the sensitive attribute $S$. In the literature, several dependence measures from statistics and information theory have been attempted to develop DP-based fair learning methodologies [Zafar et al., 2017, Mary et al., 2019, Baharlouei et al., 2019, Rezaei et al., 2020, Cho et al., 2020a,b, Lowy et al., 2022].

In practice, the applications of standard DP-based fair classification methods usually succeed in significantly reducing the DP fairness violation, while the model's original accuracy on test data can be mostly preserved. Therefore, an accuracy-based evaluation of the DP-based trained models often suggests that the improvement in the DP fairness metric can be significantly higher than the loss in the model's prediction accuracy. On the other hand, well-known studies including [Dwork et al., 2012] and [Hardt et al., 2016] have raised concerns about the potential impacts of DP-based fairness evaluation, which can disproportionately increase the inaccuracy rate among minority subgroups. To address the concerns, Hardt et al. [2016] propose and promote a different fairness notion, *equalized odds (EO)*, where the goal is a prediction variable $\widehat{Y}$ that is conditionally independent of the sensitive attribute $S$ given the true label $Y$.

In this work, we study and analyze the inductive biases of DP-based fair learning algorithms. We aim to theoreti-

cally and empirically demonstrate the biases induced by a DP-based learning framework toward the majority sensitive attribute outcome under an imbalanced distribution of the sensitive attribute over the target population. To this end, we provide theoretical results indicating the biases of DP-based fair decision rules toward the label distribution conditioned to the sensitive attribute-based majority subgroup with an occurrence probability greater than $\frac{1}{2}$. We show the existence of such a prediction distribution in a DP-based fair learning algorithm formulated by constraining the difference of demographic parity (DDP).

To reduce the biases of DP-based learning algorithms, we propose a *sensitive attribute-based distributionally robust optimization (SA-DRO)* method where the fair learner minimizes the worst-case DP-regularized loss over a set of sensitive attribute marginal distributions centered around the data-based marginal distribution. As a result, the SA-DRO approach can account for different frequencies of the sensitive attribute outcomes and thus offer a robust behavior to the changes in the sensitive attribute's majority outcome.

We present the results of several numerical experiments on the potential biases of DP-based fair classification methodologies to the sensitive attribute possessing the majority in the dataset. Our empirical findings are consistent with the theoretical results, suggesting the inductive biases of DP-based fair classification rules toward the sensitive attribute-based majority group. On the other hand, our results indicate that the SA-DRO-based fair learning method results in fair classification rules with a lower bias toward the label distribution under the majority sensitive attribute. The following is a summary of this work's main contributions:

- Analytically studying the biases of DP-based fair learning toward the majority sensitive attribute,

- Proposing a distributionally robust optimization method to lower the biases of DP-based fair classification,

- Providing numerical results on the biases of DP-based fair learning in centralized and federated learning scenarios.

## 2 RELATED WORKS

**Fairness Violation Metrics**. In this work, we focus on the learning frameworks aiming toward demographic parity (DP). Since enforcing DP to strictly hold could be costly and damaging to the learner's performance, the machine learning literature has proposed applying several metrics assessing the dependence between random variables, including: the mutual information: [Kamishima et al., 2011, Rezaei et al., 2020, Zhang et al., 2018, Cho et al., 2020a, Roh et al., 2020], Pearson correlation [Zafar et al., 2017, Beutel et al., 2019], kernel-based maximum mean discrepancy: [Prost et al., 2019], kernel density estimation of the difference of demographic parity (DDP) measures [Cho et al., 2020b], the maximal correlation [Mary et al., 2019, Baharlouei et al.,

2019, Grari et al., 2019, 2021], and the exponential Renyi mutual information [Lowy et al., 2022]. In our analysis, we mostly focus on a DP-based fair regularization scheme, while we show only weaker versions of the inductive biases could further hold in the case of mutual information and maximal correlation-based fair learning algorithms.

In addition to DP, the notions of equalized odds and equal opportunity [Hardt et al., 2016] are standard fairness notions in the literature, where the learner aims for a conditionally independent decision variable $\widehat{Y}$ of sensitive attribute $S$ given label $Y$. We note that the mentioned frameworks based on dependence measures can be aimed at equalized odds, where the dependence measure should be conditioned to label $Y$. Hence, our findings do not apply to the equalized odds fairness notion and the extension of the dependence measure-based learning algorithms aiming equalized odds.

**Fair Classification Algorithms.** Fair machine learning algorithms can be classified into three main categories: pre-processing, post-processing, and in-processing. Pre-processing algorithms [Feldman et al., 2015, Zemel et al., 2013, Calmon et al., 2017] transform biased data features into a new space where labels and sensitive attributes are statistically independent. Post-processing methods such as [Hardt et al., 2016, Pleiss et al., 2017] aim to alleviate the discriminatory impact of a classifier by modifying its ultimate decision. The focus of our work focus is only on in-processing approaches regularizing the training process toward DP-based fair models. Also, [Hashimoto et al., 2018, Wang et al., 2020, Lahoti et al., 2020] propose distributionally robust optimization (DRO) for fair classification; however, unlike our method, these works do not apply DRO on the sensitive attribute distribution to reduce the biases.

**Fairness-aware Imbalanced Learning.** To address the challenges of generalization in machine learning models, particularly when handling highly imbalanced classes and limited samples within each class, some well-known imbalanced learning methods like [Lin et al., 2017] and [Cao et al., 2019] have been proposed. More specifically, several articles [Iosifidis and Ntoutsi, 2020], [Subramanian et al., 2021], [Deng et al., 2022] and [Tarzanagh et al., 2023] extended to fairness-aware imbalanced learning dealing with imbalanced subgroups based on sensitive attributes. Compared to those methods, our SA-DRO method has more flexibility in exploring the accuracy-inductive bias trade-off controlled by varying the coefficient of the regularization term.

## 3 PRELIMINARIES

### 3.1 FAIR SUPERVISED LEARNING

To achieve fairness in supervised learning, the decision making process should not unfairly advantage or disadvantage any particular group of people based on their demographic

characteristics such as race, gender, or age, which we refer to as the sensitive attribute in this paper. In this setting, we suppose the learner has access to labeled training data $(\mathbf{x}_i, y_i, s_i)_{i=1}^n$ independently drawn from the underlying distribution $P_{\mathbf{X},Y,S}$. Here, $\mathbf{X} \in \mathcal{X} \subseteq \mathbb{R}^d$ is the $d$-dimensional feature vector, $Y \in \mathcal{Y}$ denotes the label variable, and $s \in \mathcal{S}$ denotes the sensitive attribute, which we suppose are provided for the training data.

In the supervised learning problem, the learner selects a function $f \in \mathcal{F}$ where $\mathcal{F}$ is the set of prediction functions mapping the observed $(\mathbf{X}, S)$ to the label space $\mathcal{Y}$. We use loss function $\ell : \mathcal{Y} \times \mathcal{Y} \to \mathbb{R}$ to quantify the loss $\ell(y, \widehat{y})$ when predicting $\widehat{y}$ under a true label $y$. Specifically, we consider the 0/1 loss $\ell_{0/1}(\widehat{y}, y) = \mathbf{1}(\widehat{y} \neq y)$, where $\mathbf{1}(\cdot)$ denotes the indicator function. The primary goal of the fair supervised learner is to find prediction rules $f \in \mathcal{F}$ achieving smaller values of risk function $\mathbb{E}_{(\mathbf{X},Y,S) \sim P}\big[\ell(f(\mathbf{X}, S), Y)\big]$ while having little dependence on $S$ according to the factors explained in the next subsections.

## 3.2 FAIRNESS CRITERIA

In a fair supervised learning algorithm, the learned prediction rule is expected to meet a fairness criterion. Here, we review two standard fairness criteria in the literature:

- **Demographic parity (DP)** is a fairness condition that requires the prediction $\widehat{Y}$ to be statistically independent of the sensitive attribute, $S$, i.e., for every $\widehat{y} \in \mathcal{Y}, s \in \mathcal{S}$

$$P\big(\widehat{Y} = \widehat{y} \,\big|\, S = s\big) = P\big(\widehat{Y} = \widehat{y}\big)$$

where $\widehat{Y} = f(\mathbf{X}, S)$ represents the predicted label. A standard quantification of the violation of DP is the Difference of Demographic Parity (DDP):

$$\text{DDP}(\widehat{Y}, S) = \sum_{y \in \mathcal{Y}, s \in \mathcal{S}} \Big| P\big(\widehat{Y} = y | S = s\big) - P\big(\widehat{Y} = y\big) \Big|$$

- **Equalized Odds (EO)** [Hardt et al., 2016] is a fairness condition requiring the predicted label $Y$ to be conditionally independent from sensitive attribute $S$ given actual label $Y$, i.e. for every $s \in \mathcal{S}, y, \widehat{y} \in \mathcal{Y}$

$$P\big(\widehat{Y} = \widehat{y} \,\big|\, Y = y, S = s\big) = P\big(\widehat{Y} = \widehat{y} \,\big|\, Y = y\big).$$

A sensible measurement of the lack of EO is the Difference of Equalized Odds (DEO):

$$\text{DEO}(\widehat{Y}, S | Y) = \sum_{s \in \mathcal{S}, y, \hat{y} \in \mathcal{Y}} \Big| P\big(\widehat{Y} = \widehat{y} \,\big|\, Y = y, S = s\big)$$
$$- P\big(\widehat{Y} = \widehat{y} \,\big|\, Y = y\big) \Big|$$

## 3.3 DEPENDENCE MEASURES FOR FAIR SUPERVISED LEARNING

To measure the DP-based fairness violation, the machine learning literature has proposed the application of several dependence measures which we analyze in the paper. In the following, we review some of the applied dependence metrics:

- **Mutual Information (MI)**: Mutual information $I(Y; S)$ is a standard measure of the dependence between random variables $Y$ and $S$ used for developing fair learning methods [Cho et al., 2020a]. The mutual information $I(Y; S)$ is defined as

$$I(Y; S) := \sum_{y \in \mathcal{Y}, s \in \mathcal{S}} P_{Y,S}(y, s) \log \frac{P_{Y,S}(y, s)}{P_Y(y) P_S(s)}$$

It can be seen that $I(Y; S) = \mathrm{D}_{\mathrm{KL}}(P_{Y,S}; P_Y P_S)$ is the KL-divergence between joint distribution $P_{Y,S}$ and product of marginal distributions $P_Y \times P_S$, implying $I(Y; S) = 0$ if and only if $Y$ and $S$ are statistically independent, i.e., $Y \perp S$. Note that KL-divergence is a special case of $f$-divergence $d_f(P, Q) = \mathbb{E}_P[f(P(x)/Q(x))]$ with $f(t) = t \log t$.

- **Maximal Correlation (MC)**: The maximal correlation $\rho_m(Y, S)$ is the maximum Pearson correlation $\rho_P\big(f(Y), g(S)\big) = \frac{\mathrm{Cov}(f(Y), g(S))}{\sqrt{\mathrm{Var}(f(Y)) \mathrm{Var}(g(S))}}$ between $f(Y)$ and $g(S)$ over all functions $f$, $g$. The maximal correlation can be simplified to the optimal value of the following optimization:

$$\rho_m(Y, S) := \sup_{\substack{f,g: \, \mathbb{E}[f(Y)] = \mathbb{E}[g(S)] = 0 \\ \mathbb{E}[f^2(Y)] = \mathbb{E}[g^2(S)] = 1}} \mathbb{E}\big[f(Y) g(S)\big]$$

Maximal correlation has been utilized as a measure of demographic parity in the literature on fair learning algorithms [Mary et al., 2019, Baharlouei et al., 2019].

- **Exponential Rényi Mutual Information (ERMI)**: The ERMI between random variables $Y$ and $S$, which is considered by Lowy et al. [2022] as the dependence measure of fairness penalty, is

$$\rho_E(Y, S) := \chi^2(P_{Y,S}; P_Y \times P_S),$$

i.e, the $\chi^2$-divergence between the joint distribution $P_{Y,S}$ and the product of marginal distributions $P_Y \times P_S$. Similar to KL-divergence, $\chi^2$-divergence is an $f$-divergence $d_f(P, Q)$ with $f(t) = (t - 1)^2$. Similar to the previous two dependence measures, $\rho_E(Y, S) = 0$ if and only if $Y$, $S$ are independent.

# 4 INDUCTIVE BIASES OF DP-BASED FAIR SUPERVISED LEARNING

As discussed earlier, fair learning based on the demographic parity (DP) notion requires a bounded dependence between

the classifier's output $\widehat{Y}$ and sensitive attribute $S$. A standard approach widely-used in the literature to DP-based fair classification is to target the following optimization problem for a dependence measure $\rho(\widehat{Y}, S)$ between $S$ and predicted variable $\widehat{Y} = f(\mathbf{X}, S)$ given a randomized prediction rule $f \in \mathcal{F}$ where $\mathcal{F}$ is a set of functions mapping $\mathbf{x} \in \mathcal{X}, s \in \mathcal{S}$ to a random $\widehat{Y} \in \mathcal{Y}$ with a conditional distribution $P_{\widehat{Y}|\mathbf{X},S}$:

$$\min_{f \in \mathcal{F}} \quad \mathbb{E}_{p_{\mathbf{X},Y,S}}\left[\ell_{0/1}(\widehat{Y}, Y)\right] \qquad (1)$$
$$\text{subject to} \quad \rho(\widehat{Y}, S) \leq \epsilon$$

Our first theorem shows that if one chooses DDP as the dependence measure $\rho$ and that $Y$ can be deterministically determined by $\mathbf{X}, S$, then for the optimal solution $\widehat{Y} = f^*(\mathbf{X}, S)$ to the above problem, the conditional distribution $P_{\widehat{Y}|S=s}$ for every $s$ will be close to the conditional distribution $P_{Y|S=s_{\max}}$ of $Y$ conditioned on the majority sensitive attribute $s_{\max} = \arg\max_{s \in \mathcal{S}} P_S(s)$. In the theorem, we use TV to denote the total variation distance between distributions $P_Y$ and $Q_Y$ defined as

$$TV(P_Y, Q_Y) := \frac{1}{2} \sum_{y \in \mathcal{Y}} |P_Y(y) - Q_Y(y)|$$

**Theorem 1.** *Consider fair learning problem* (1) *where $\rho$ is the DDP function and $\mathcal{F}$ is the space of all randomized maps generating all conditional distribution $P_{\widehat{Y}|\mathbf{X},S}$'s. Suppose that $Y = h(\mathbf{X}, S)$ is a deterministic function $h$ of $\mathbf{X}, S$. Then, if the majority sensitive attribute $s_{\max}$ satisfies $P(S = s_{\max}) = \frac{1}{2} + \delta$ for a positive $\delta > 0$, then the following bound holds for the optimal predicted variable $\widehat{Y} = f^*(\mathbf{X}, S)$ where $f^*$ is the optimal solution to* (1)

$$\forall s \in \mathcal{S}: \quad TV\left(P_{\widehat{Y}|S=s}, P_{Y|S=s_{\max}}\right) \leq \left(\frac{1}{2} + \frac{1}{4\delta}\right)\epsilon$$

*Proof.* We defer the proof to the Appendix. $\quad\square$

**Corollary 1.** *In the setting of Theorem 1, if $\epsilon = 0$, i.e., $\widehat{Y}$ and $S$ are constrained to be statistically independent, then $P(S = s_{\max}) > \frac{1}{2}$ results in the following for the optimal predicted variable $\widehat{Y} = f^*(\mathbf{X}, S)$:*

$$\forall s \in \mathcal{S}: \quad P_{\widehat{Y}|S=s} = P_{Y|S=s_{\max}}$$

The above results show that given a sensitive attribute $s_{\max}$ holding more than half of the training data, the optimal DDP-fair prediction $\widehat{Y}$ will possess a conditional distribution $P_{\widehat{Y}|S=s}$ which for every $s$ is at a bounded TV-distance from the majority $s_{\max}$-based conditional distribution $P_{Y|S=s_{\max}}$. Therefore, the results indicate the inductive bias of a DDP-based fair learning toward the majority sensitive attribute. Next, we show that a weaker version of the DDP-based bias could also hold for the mutual information, ERMI, and maximal correlation-based fair learning.

**Theorem 2.** *Consider the fair learning setting in Theorem 1 with a different selection of dependence measure $\rho$. Then,*

- *assuming $\rho(\widehat{Y}, S)$ is the mutual information $I(\widehat{Y}; S)$:*

$$\mathbb{E}_{s \sim P_S}\left[TV\left(P_{\widehat{Y}|S=s}, P_{Y|S=s_{\max}}\right)\right] \leq \left(\frac{1}{2} + \frac{1}{4\delta}\right)\sqrt{\frac{2\epsilon}{\log e}}$$

- *assuming $\rho(\widehat{Y}, S)$ is the ERMI $\rho_E(\widehat{Y}, S)$ and defining $u(\epsilon) = \max\{\epsilon, \sqrt{\epsilon}\}$:*

$$\mathbb{E}_{s \sim P_S}\left[TV\left(P_{\widehat{Y}|S=s}, P_{Y|S=s_{\max}}\right)\right] \leq \left(\frac{1}{2} + \frac{1}{4\delta}\right)u(\epsilon)$$

- *assuming $\rho(\widehat{Y}, S)$ is maximal correlation $\rho_m(\widehat{Y}, S)$ and $r = \min\{|\mathcal{S}|, |\mathcal{Y}|\} - 1$ ($|\cdot|$ denotes a set's cardinality):*

$$\mathbb{E}_{s \sim P_S}\left[TV\left(P_{\widehat{Y}|S=s}, P_{Y|S=s_{\max}}\right)\right] \leq \left(\frac{1}{2} + \frac{1}{4\delta}\right)u(r\epsilon)$$

*Proof.* We defer the proof to the Appendix. $\quad\square$

We remark the difference between the bias levels shown for the DDP case in Theorem 1 and the other dependence metrics in Theorem 2. The bias level for a DDP-based fair learner could be considerably stronger than that of mutual information, ERMI, and maximal correlation-based fair learners, as the wort-case of total variations in Theorem 1 is replaced by their expectation according to $P_S$ in Theorem 2.

## 4.1 EXTENDING THE THEORETICAL RESULTS TO RANDOMIZED PREDICTION RULES

Here, we consider the possibility of a randomized mapping from $(\mathbf{X}, S)$ to $Y$. Such a possibility needs to be considered when the actual label $Y$ may not be deterministically determined by $\mathbf{X}, S$. Therefore, we formulate and analyze the following generalization of the problem formulation in (1) where we attempt to find the conditional distribution $P_{Y|\mathbf{X},S}$:

$$\min_{Q_{\widehat{Y}|\mathbf{X},S} \in \mathcal{Q}} \mathbb{E}_{P_{\mathbf{X},S}}\left[\ell_{TV}\left(Q_{\widehat{Y}|\mathbf{X}=\mathbf{x},S=s}, P_{Y|\mathbf{X}=\mathbf{x},S=s}\right)\right] \quad (2)$$
$$\text{subject to} \quad \rho(\widehat{Y}, S) \leq \epsilon$$

In this formulation, we aim to find an accurate estimation of the conditional distribution $Q_{\widehat{Y}|\mathbf{X},S}$ from a feasible set $\mathcal{Q}$ which corresponds to the function set $\mathcal{F}$ in (1). We measure the learning performance under every outcome $\mathbf{x}, s \sim P_{\mathbf{X},S}$ using the total variation loss $\ell_{TV}(P, Q) = TV(P, Q)$. Note that the total variation loss generalizes the 0/1 loss to the space of probability measures, since it is the minimum expected 0/1 loss under the optimal coupling between the marginal distributions:

$$\ell_{\mathrm{TV}}(P, Q) = \min_{\substack{M_{\widehat{Y},Y}: M_{\widehat{Y}}=P \\ M_Y=Q}} \mathbb{E}_M\left[\ell_{0/1}(\widehat{Y}, Y)\right].$$

Therefore, if under both $Q$ and $P$, $Y$ is determined deterministically by $\mathbf{X}, S$, the above TV-loss will be the same as the expected 0/1 loss of the deterministic classification rule following such $Q_{\widehat{Y}|\mathbf{X},S}$. In the following theorem, we attempt to relax the assumptions in Theorems 1-2 to apply them to learning settings where $Y$ may not be completely determined by $\mathbf{X}, S$.

**Theorem 3.** *Consider the settings in Theorem 1 and Theorem 2 where we instead consider the generalized formulation* (2) *and do not require that $Y$ is a function of $\mathbf{X}, S$. Suppose a function $\phi : \mathcal{X} \times \mathcal{Y} \to \mathbb{R}$ exists such that the underlying distribution $P_{\mathbf{X},Y,S}$ satisfies the following property on the ratio between conditional distributions $P_{\mathbf{X}|Y,S}$ and $P_{\mathbf{X}|S}$:*

$$\forall \mathbf{x} \in \mathcal{X}, y \in \mathcal{Y}, s \in \mathcal{S} : \quad \frac{P(\mathbf{x} \mid y, s)}{P(\mathbf{x} \mid s)} = \phi(\mathbf{x}, y). \quad (3)$$

*Then, the conclusions in Theorems 1,2 will remain valid.*

*Proof.* We defer the proof to the Appendix. □

**Remark 1.** *Note that the assumption in the above theorem is equivalent to a $s$-independent ratio $\frac{P(\mathbf{x}|y,s)}{P(\mathbf{x}|s)}$. In particular, this assumption will hold if the random vector $\mathbf{X}$ can be decomposed to $\left[ g(S), \widetilde{\mathbf{X}} \right]$, where $g$ is a deterministic function, and under the true distribution $p_{\mathbf{X},Y,S}$, $\widetilde{\mathbf{X}}$ satisfies $\widetilde{\mathbf{X}} \perp S$, i.e., is independent from $S$, and $\widetilde{\mathbf{X}} \perp S \mid Y$, i.e, $\widetilde{\mathbf{X}}$ is conditionally independent from $S$ given $Y$.*

Finally, we attempt to further relax the assumption in Theorem 3 when the distribution ratio $P(\mathbf{x}|y,s)/P(\mathbf{x}|s)$ may not be completely independent of the outcome $S = s$. The next theorem shows a quantification of the deviation from the assumption and how much it can impact the result.

**Theorem 4.** *Consider the setting of Theorem 3 and the formulation* (2). *We consider the TV-based dependence $\rho_{TV}(Y, S) := \mathbb{E}_{s \sim P_S} \left[ \text{TV}(P_{Y|S=s}, P_Y) \right]$ in the problem. Suppose for functions $\phi_L, \phi_U : \mathcal{X} \times \mathcal{Y} \to \mathbb{R}$, the following holds for every $\mathbf{x} \in \mathcal{X}, y \in \mathcal{Y}, s \in \mathcal{S}$:*

$$\phi_L(\mathbf{x}, y) \leq \frac{p(\mathbf{x}|y,s)}{p(\mathbf{x}|s)} \leq \phi_U(\mathbf{x}, y).$$

*Define $\Delta(\mathbf{x}, y) = \phi_U(\mathbf{x}, y) - \phi_L(\mathbf{x}, y)$. Then, if $\frac{\epsilon}{2} \geq \mathbb{E}_{P_X P_{Y|S=s_{\max}}} \left[ \Delta(\mathbf{x}, y) \right]$, for the optimal $Q^*_{\widehat{Y}|\mathbf{X},S}$, $P^*_{\widehat{Y},\mathbf{X},S} = Q^*_{\widehat{Y}|\mathbf{X},S} \cdot P_{\mathbf{X},S}$ satisfies*

$$\mathbb{E}_{s \sim P_S} \left[ \text{TV}\left( P_{\widehat{Y}|S=s}, P_{Y|S=s_{\max}} \right) \right] \leq 2\epsilon \left( 1 + \frac{1}{2\delta} \right)$$

*Proof.* We defer the proof to the Appendix. □

---

**Algorithm 1** Sensitive Attribute-based Distributionally Robust Optimization (SA-DRO) Fair Learning Algorithm

1: **Input:** Training data $\{(\mathbf{x}_i, y_i, s_i)_{i=1}^n\}$, parameters $\lambda, \delta \geq 0$, divergence $d$, dependence measure $\rho$, stepsizes $\alpha_w, \alpha_q > 0$, running iterations $T > 0$
2: **Initialize** classifier weight $\mathbf{w}$ and distribution $\mathbf{q} = \mathbf{p}_s$
3: **for** t $\in \{1, \ldots, T\}$ **do**
4:     Compute weight gradient of the classifier $f_\mathbf{w}$:
$$\mathbf{g_w} = \sum_{i=1}^n \left[ \frac{q_{s_i}}{n} \nabla_\mathbf{w} \ell\big(f_\mathbf{w}(\mathbf{x}_i), y_i\big) \right] + \lambda \nabla_\mathbf{w} \rho\big(f_\mathbf{w}(\mathbf{X}), S\big)$$
5:     Update $\mathbf{w}$ with gradient descent: $\mathbf{w} \leftarrow \mathbf{w} - \alpha_w \mathbf{g_w}$
6:     Compute the gradient of $q_s$ for every $s \in \mathcal{S}$:
$$g_{\mathbf{q}_s} = \frac{1}{n} \sum_{i:s_i=s} \left[ \ell\big(f_\mathbf{w}(\mathbf{x}_i), y_i\big) \right] + \lambda \frac{\partial \rho\big(f_\mathbf{w}(\mathbf{x}_{1:n}), s_{1:n}\big)}{\partial q_s}$$
7:     Update $\mathbf{q}$ with projected gradient ascent:
    $\mathbf{q} \leftarrow \Pi_{\{\mathbf{q}: \, d(\mathbf{q}, \mathbf{p}_s)) \leq \delta\}} \big( \mathbf{q} + \alpha_q g_\mathbf{q} \big)$
8: **end for**

---

# 5 A DISTRIBUTIONALLY ROBUST OPTIMIZATION APPROACH TO DP-BASED FAIR LEARNING

In this section, we propose a distributionally robust optimization method to reduce the biases of DP-based fair learning algorithms toward the majority sensitive attribute. As discussed before, the optimization of the original risk function under the true distribution $p_{\mathbf{X},Y,S}$ would lead to biases if a sensitive attribute $s_{\max}$ occurs considerably more than half of the times. To shield the learning algorithm against such biases, we propose applying distributionally robust optimization (DRO) and consider the worst-case expected 0/1 loss over a distribution ball around the sensitive attribute distribution $p_S$ as the target in the learning problem. This approach leads to the *sensitive attribute-based distributionally robust optimization (SA-DRO)* algorithm solving the following formulation of the fair learning problem with dependence metric $\rho(\widehat{Y}, S)$:

$$\min_{f \in \mathcal{F}} \max_{Q_S : d(Q_S, P_S) \leq \delta} \mathbb{E}_{P_{\mathbf{X},Y|S} \cdot Q_S} \left[ \ell_{0/1}(\widehat{Y}, Y) \right] + \lambda \rho(\widehat{Y}, S) \tag{4}$$

According to this formulation, we solve the Lagrangian version of optimization problem (1) when $S$'s marginal distribution $q_S$ leads to the worst-case fair-regularized risk function in a distribution ball $\{ q_S : d(q_S, p_S) \leq \delta \}$ where $d$ is a distance measure between probability distributions. In this formulation, we consider assigning different weights to samples with different sensitive attributes, which may result in different majority sensitive attributes. Since we are optimizing the worst-case performance over the distribution

ball with a $\delta$ radius, the inductive biases discussed in the previous would become less effective under a greater $\delta$.

The proposed SA-DRO formulation results in Algorithm 1 which applies projected gradient descent ascent (GDA) to solve the minimax optimization problem in (4). Here, we use a parameterized classifier $f_{\mathbf{w}}$ to apply a gradient-based training algorithm. Also, the distance $d$ can be chosen as any standard $f$-divergence. In our experiments, we attempted the $\chi^2$-divergence divergence, which has been well-explored in the literature [Namkoong and Duchi, 2016, Bertsimas et al., 2019, Rahimian and Mehrotra, 2019]. Furthermore, a Lagrangian form of the SA-DRO problem (4) can be considered where the DRO constraint on $Q_S$ is transferred to the inner maximization objective function as $-\zeta d(Q_S, P_S)$ for a Lagrangian coefficient $\zeta > 0$.

# 6 NUMERICAL RESULTS

## 6.1 EXPERIMENTAL SETUP

**Datasets.** In our experiments, we attempted the following standard datasets in the machine learning literature:

1. *COMPAS* dataset with 12 features and a binary label on whether a subject has recidivism in two years, where the sensitive attribute is the binary race feature[1]. To simulate a setting with imbalanced sensitive attribute distribution, we considered 2500 training and 750 test samples, in both of which 80% are from $S = 0$ "non-Caucasian" and 20% of the samples are from $S = 1$ "Caucasian".

2. *Adult* dataset with 64 binary features and a binary label indicating whether a person has more than 50K annual income. In this case, gender is considered as the sensitive attribute[2]. In our experiments, we used 15k training and 5k test samples, where, to simulate an imbalanced distribution on the sensitive attribute, 80% of the data have male gender and 20% of the samples are females.

3. *CelebA* Proposed by [Liu et al., 2018], containing the pictures of celebrities with 40 attribute annotations, where we considered "gender" as a binary label, and the sensitive attribute is the binary variable on blond/non-blond hair. In the experiments, we used 5k training samples and 2k test samples. To simulate an imbalanced sensitive attribute distribution, 80% of both training and test samples are marked with Blond hair and 20% samples are marked with non-blond hair.

**DP-based Learning Methods**: We performed the experiments using the following DP-based fair classification methods: 1) DDP-based KDE method [Cho et al., 2020a] and FACL [Mary et al., 2019], 2) the mutual information-based fair classifier [Cho et al., 2020b], 3) the maximal

Correlation-based RFI classifier [Baharlouei et al., 2019], to learn binary classification models on COMPAS and Adult datasets. For CelebA experiments, we used the following two DP-based fair classification methods: KDE method [Cho et al., 2020a], and mutual information (MI) fair classifier [Cho et al., 2020b].

In the experiments, we attempted both a logistic regression classifier with a linear prediction model and a neural net classifier. The neural net architecture was 1) for the COMPAS case, a multi-layer perceptron (MLP) with 2 hidden layers with 128 neurons per layer, 2) for the Adult case, an MLP with 4 hidden layers with 512 neurons per layer, 3) for the CelebA case, the ResNet-18 [He et al., 2016] architecture suited for the image input in the experiments.

**Evaluation criteria**: To evaluate the trained models, we used the averaged accuracy rate (Acc) as the classification performance metric and the Difference of Demographic Parity (DDP) as the fairness metric. Moreover, to quantify the bias effects of fair learners, we measured the *negative rate (NR) conditioned to a sensitive attribute* defined as $\mathrm{NR}(s) := P(\hat{Y} = 0 \mid S = s)$. This metric is defined to quantify the variations in prediction outcomes across subgroups with different sensitive attribute values.

## 6.2 INDUCTIVE BIASES OF MODELS TRAINED IN DP-BASED FAIR LEARNING

To numerically analyze the effects of DP-based fair classification algorithms, we varied the regularization penalty coefficient $\lambda$ over the range $[0, 1]$. Note that $\lambda = 0$ means an ERM setting with no fairness constraint, while $\lambda = 1$ is the strongest fairness regularization coefficient over the range $[0, 1]$.

As the evaluated accuracy and DDP values in Figures 1 indicate, the DP-based fair learning algorithms managed to significantly reduce the DDP fairness violation while compromising less than 2% in accuracy. On the other hand, the negative rate ($\hat{Y} = 0$ prediction rate) across the two outcomes $S = 0$, $S = 1$ of the sensitive attribute tend toward the majority sensitive attribute as the DP-based fairness regularization became stronger, suggesting the conditional distribution of the prediction $\hat{Y}$ given different sensitive attribute outcome $S = s$'s moved closer to that of the majority sensitive attribute. The observed behavior held similarly using both the linear logistic regression model in Figure 2 and Figure 3, and Figure 4 shows how the inductive biases lead to misclassification on CelebA dataset.

**DRO-based Fair Learning.** We tested Algorithm 1 utilizing a sensitive attribute-based distributional robust optimization (SA-DRO) to DP-based fair learning algorithms. In our experiments, we applied the SA-DRO algorithm to the DDP-based KDE fair learning algorithm proposed by Cho et al. [2020b], and RFI proposed by Baharlouei et al. [2019].

---

[1]https://github.com/propublica/compas-analysis

[2]https://archive.ics.uci.edu/dataset/2/adult

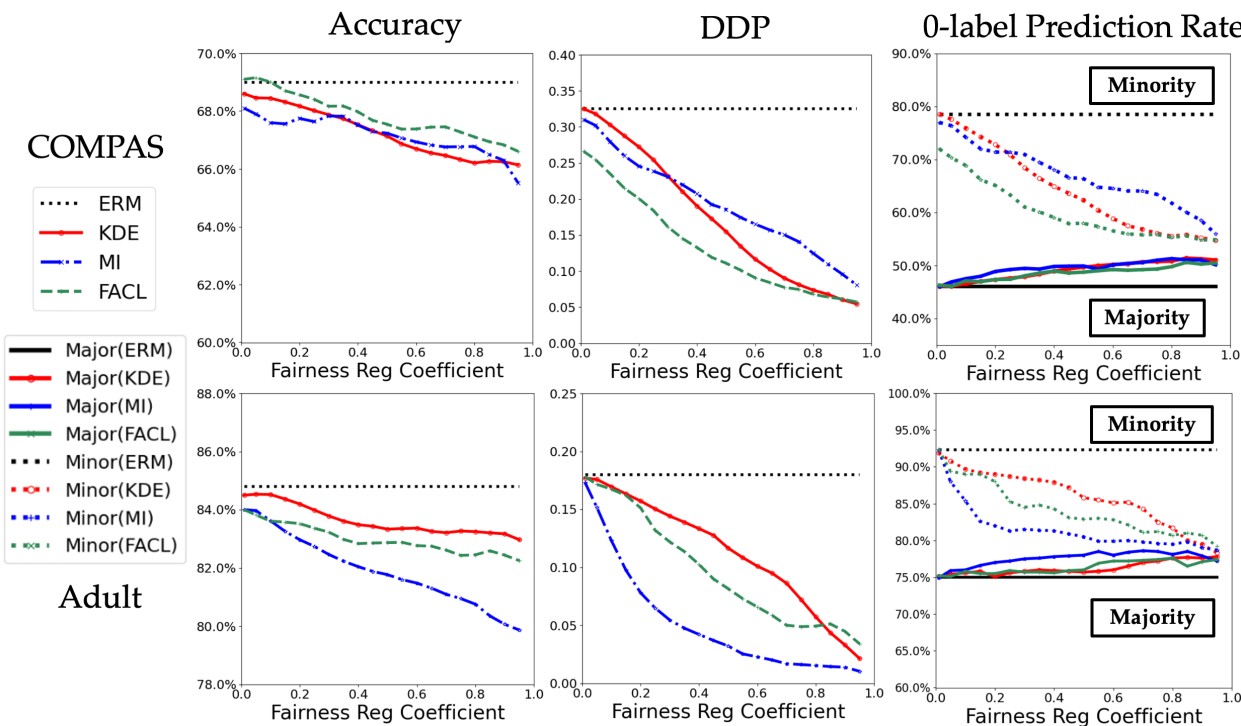

Figure 1: The first two columns show the trade-off between accuracy and DDP on the COMPAS and Adult dataset by applying NN-based fair classification methods, while the third column shows that the $\mathrm{NR}(s)$ for each subgroup $s \in \{0, 1\}$ will converge to near the majority sensitive attribute.

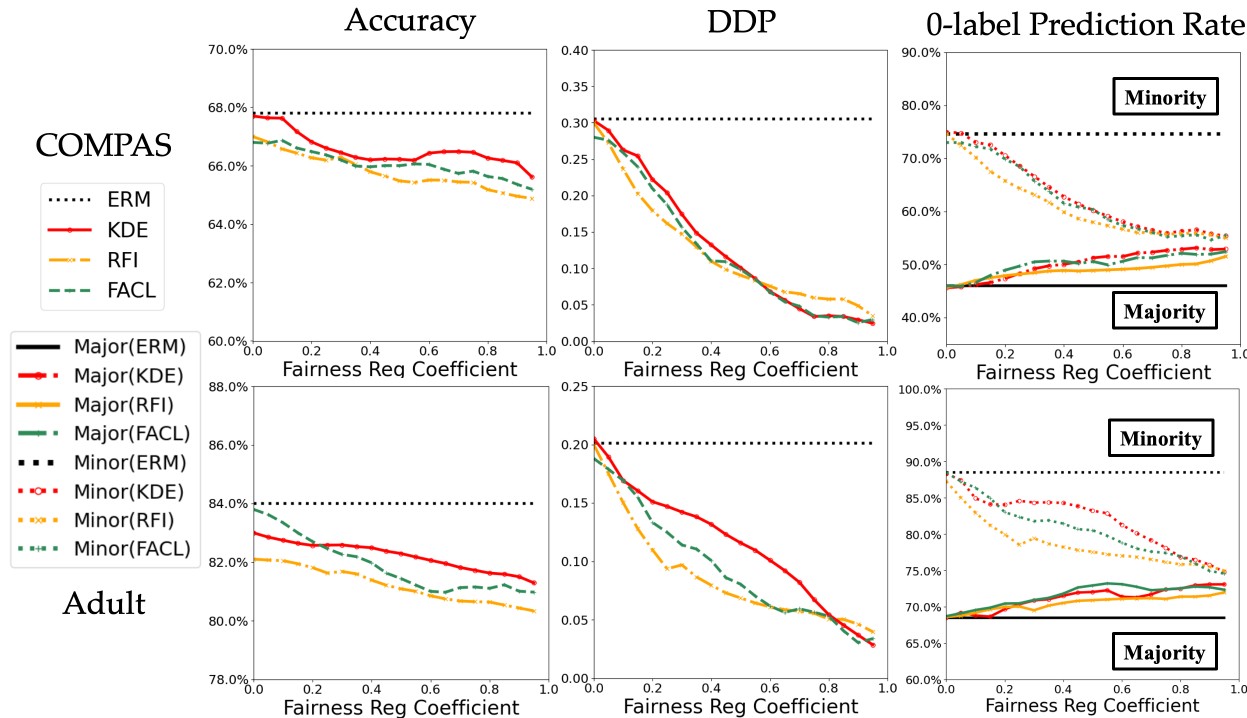

Figure 2: The first two columns show the trade-off between accuracy and DDP on the COMPAS and Adult dataset by applying LR-based fair classification methods, while the third column shows that the $\mathrm{NR}(s)$ for each subgroup $s \in \{0, 1\}$ will converge to near the majority sensitive attribute.

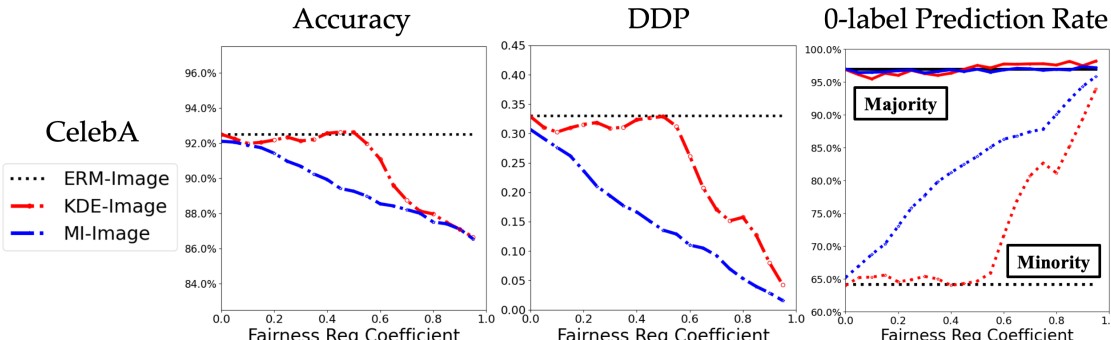

Figure 3: Both (a) and (b) show the trade-off between accuracy and DDP on the imbalanced CelebA dataset by applying MI fair classification method, while (c) shows that the $NR(s)$ for each subgroup will converge to the majority, thus causing more discrimination on the minority group.

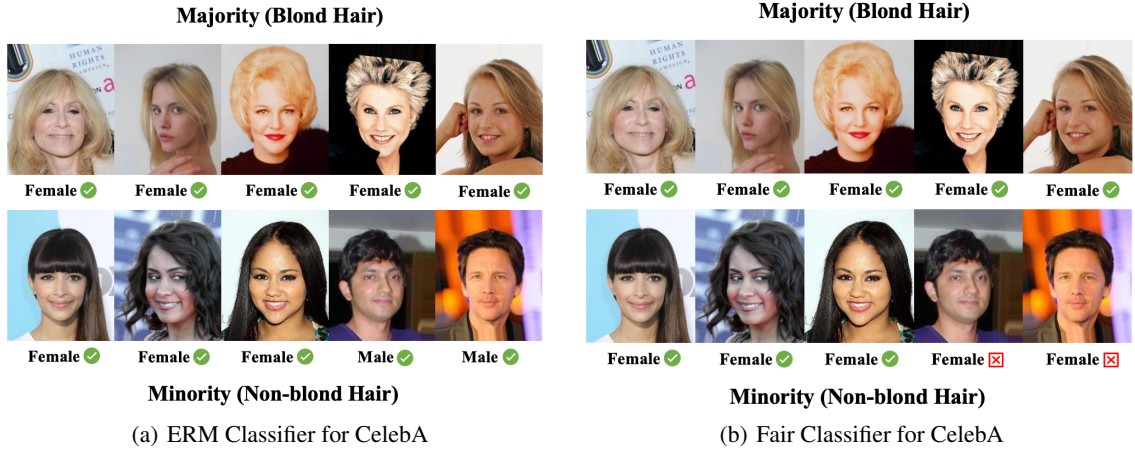

(a) ERM Classifier for CelebA        (b) Fair Classifier for CelebA

Figure 4: Blond hair samples (Majority, Upper) and Non-blond hair samples (Minority, Lower) in CelebA Dataset predicted by ERM(NN) and MI respectively. The results show that the model has 57.3% and 98.8% negative rates, i.e. prefers to predict all samples being female in Minority, even maintaining almost the same level of accuracy in the whole group.

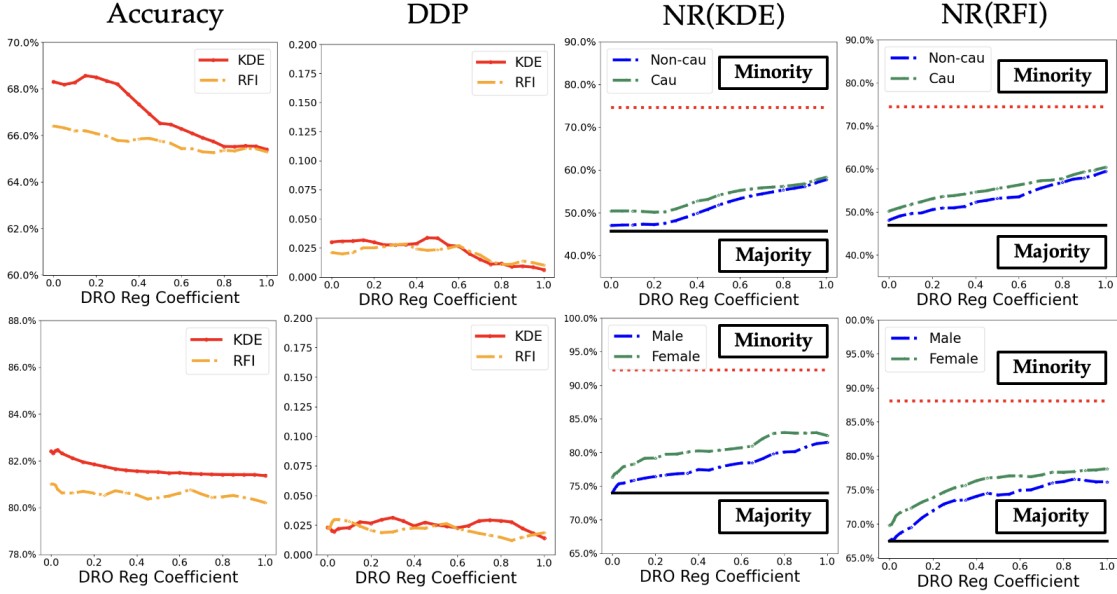

Figure 5: Accuracy, DDP, and $NR(s)$ values attained by SA-DRO while varying the Lagrangian coefficient of the DRO regularization term on COMPAS (upper) and Adult (lower) datasets.

Table 1: Numerical Results on COMPAS and Adult, non-DRO vs SA-DRO implementations.

| | Method | Acc($\uparrow$) | DDP$\downarrow$ | NR($s=0$) | NR($s=1$) |
|---|---|---|---|---|---|
| COMPAS | ERM(NN) | 68.0% | 0.287 | 46.0% | 74.7% |
| | KDE | 66.8% | 0.027 | 46.3% | 49.0% |
| | KDE (SA-DRO) | 66.0% | 0.009 | 61.6% | 62.5% |
| | ERM(LR) | 67.5% | 0.287 | 47.0% | 74.5% |
| | RFI | 66.4% | 0.021 | 48.1% | 50.2% |
| | RFI (SA-DRO) | 65.4% | 0.017 | 59.3% | 61.0% |
| Adult | ERM(NN) | 85.1% | 0.183 | 92.3% | 74.0% |
| | KDE | 83.2% | 0.023 | 77.3% | 75.0% |
| | KDE (SA-DRO) | 82.5% | 0.012 | 84.6% | 83.4% |
| | ERM(LR) | 82.0% | 0.189 | 88.1% | 67.5% |
| | RFI | 80.6% | 0.019 | 69.8% | 67.9% |
| | RFI (SA-DRO) | 80.1% | 0.021 | 78.3% | 76.2% |

We kept the fairness regularization penalty coefficient to be $\lambda = 0.9$. Following the commonly-used implementation of DRO, we used a Lagrangian penalty term $-\zeta d(P_S, Q_S)$ in the inner maximization problem to perform DRO. Therefore, the DRO regularization coefficient, also the Lagrangian multiplier $\zeta$, can take over the range $[0, +\infty]$, in the table 1, we set $\zeta = 0.9$ for SA-DRO case. The visualized results for various DRO regularization coefficients can be found in Appendix.

As Table 1 shows, we observed that the proposed SA-DRO reduces the tendency of the fair learning algorithm toward the majority sensitive attribute, and the resulting negative prediction rates conditioned to sensitive attribute outcomes became closer to the midpoint between the majority and minority conditional accuracies. On the other hand, the SA-DRO-based algorithms still achieve a low DDP value while the accuracy drop is less than 1%.

In Figure 5, we visualized the results of applying SA-DRO algorithm to the DP-based KDE by Cho et al. [2020b], and RFI by Baharlouei et al. [2019] for various DRO coefficients. We kept the fairness regularization penalty coefficient to be $\lambda = 0.9$, and the DRO regularization coefficient took over the range $[0, 1]$. This Figure 5 shows that the accuracy and DDP among the whole groups or different subgroups are slightly affected, while the NR($s$) for different subgroups will shift from the majority group to the midpoint between the minority group and the majority group to effectively reduce the inductive biases.

### 6.3 DP-BASED FAIR CLASSIFICATION IN HETEROGENEOUS FEDERATED LEARNING

To numerically show the implications of the inductive biases of DP-based fair learning algorithms, we simulated a heterogeneous federated learning setting with multiple clients where the sensitive attribute has different distributions across clients. To do this, we split the Adult dataset into 4 subsets of 3k samples to be distributed among 4 clients in the federated learning. While 80% of the training data in Client 1 (minority subgroup in the network) had Female as sensitive attribute, only 20% of Clients 2-4 were female samples. We used the same male/female data proportion to assign 750 test samples to the clients.

For the baseline federated learning method with no fairness regularization, we utilized the FedAvg algorithm [McMahan et al., 2017]. For the DP-based fair federated learning algorithms, we attempted the DDP-based KDE and FACL algorithms which result in single-level optimization problem and hence can be optimized in a distributed learning problem by averaging as in FedAvg. We refer to the extended federated learning version of these algorithms as FedKDE and FedFACL. We also tested our SA-DRO implementations of FedKDE and FedFACL, as well as the localized ERM, KDE, FACL models where each client trained a separate model only on her own data.

To show the impacts of such inductive biases in practice, we focused on a setting with heterogeneous sensitive attribute distributions across clients where the clients' majority sensitive attribute outcome may not agree. Figure 6 illustrates such a federated learning scenario over the Adult dataset, where Client 1's majority sensitive attribute (female samples) is different from the network's majority group (male samples). In the experiment, Client 1's test accuracy with a DP-based fair federated learning was significantly lower than the test accuracy of a locally-trained fair model learned only on Client 1's data. Such numerical results suggest the possibility of the minority clients' lack of incentive to participate in the fair federated learning process.

To test our proposed DRO approach, we applied the SA-DRO method. As our numerical results in Table 2 indicate, the inductive biases of DP-based federated learning could considerably lower the accuracy of Client 1 with a different majority sensitive attribute compared to the other clients. The accuracy drop led to a lower accuracy compared to Client 1's locally fair trained model without any collaboration with the other clients, which may affect the client's incentive to participate in the federated learning process. On the other hand, the SA-DRO implementations of the KDE and FACL methods achieved a better accuracy than Client 1's local model while preserving the accuracy for the majority clients and maintaining the same level of DDP no more than 0.05. We found similar results in the CelebA federated learning experiments, as in Table 3 in the Appendix.

## 7 CONCLUSION

In this work, we attempted to demonstrate the inductive biases of in-processing fair learning algorithms aiming to

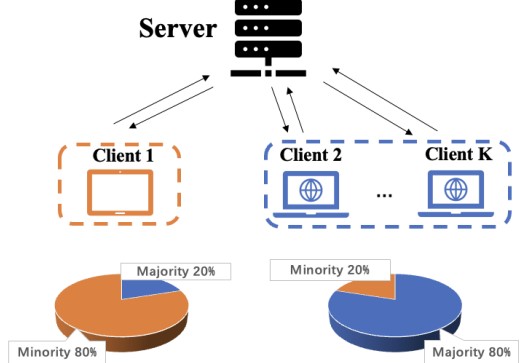

| | Minority Client | | Majority Clients | |
| --- | --- | --- | --- | --- |
| | Accuracy Acc($\uparrow$) | Fairness Violation DDP($\downarrow$) | Accuracy Acc($\uparrow$) | Fairness Violation DDP($\downarrow$) |
| Standard (No Fair Reg.) Federated Training | 82.5% | 0.208 | 90.3% | 0.206 |
| Standard (No Fair Reg.) Localized Training | 81.6% | 0.203 | 89.0% | 0.246 |
| DP-based (With Fair Reg.) Federated Training | 74.8% | 0.022 | 90.0% | 0.029 |
| DP-based (With Fair Reg.) Localized Training | 79.0% | 0.032 | 88.2% | 0.014 |

Figure 6: Biases of DP-based learning algorithms in federated learning with heterogeneous sensitive attribute distributions: 80% of the training data in Client 1 comes from the minority subgroup (female) of the entire network, while the other clients have 20% of their data from the minority subgroup. The DP-based KDE fair federated learning algorithm led to a significantly lower accuracy for Client 1 compared to the test accuracy of Client 1's locally (non-federated) trained model.

Table 2: Accuracy and DDP on Adult dataset

| | Client 1 (Minority) | | Client 2-4 (Majority) | |
| --- | --- | --- | --- | --- |
| | Acc($\uparrow$) | DDP($\downarrow$) | Acc($\uparrow$) | DDP($\downarrow$) |
| FedAvg | 82.5% | 0.208 | 90.3% | 0.206 |
| ERM(Local) | 81.6% | 0.203 | 89.0% | 0.246 |
| FedKDE | 74.8% | 0.022 | 89.9% | 0.029 |
| FedFACL | 74.5% | 0.014 | 89.7% | 0.031 |
| **SA-DRO-FedKDE** | **79.3%** | 0.041 | 89.6% | 0.042 |
| **SA-DRO-FedFACL** | **79.0%** | 0.049 | 89.1% | 0.036 |
| KDE(Local) | 79.0% | 0.032 | 88.2% | 0.014 |
| FACL(Local) | 79.1% | 0.025 | 88.6% | 0.017 |

achieve demographic parity (DP). We also proposed a distributionally robust optimization scheme to reduce the biases toward the majority sensitive attribute. An interesting future direction to our work is to search for similar biases in pre-processing and post-processing fair learning methods. Also, the theoretical comparison between different dependence measures such as mutual information, Pearson correlation, and the maximal correlation on the inductive bias levels will be an interesting topic for future exploration. Finally, characterizing the trade-off between accuracy, fairness violation, and biases toward the majority subgroups will help to better understand the costs of DP-based fair learning.

## LIMITATIONS AND BROADER IMPACT

Our theoretical analysis focuses on the total variation loss, which can limit its application to other popular loss functions in statistical learning, e.g. the cross entropy loss. Extending the analytical findings on the inductive biases of fair learning algorithms to other loss functions will be a future direction. Also, we clarify that due to the relatively high

dimensions of the datasets in our numerical experiments, we were unable to validate the assumption in Theorems 3, 4 in the experiments. However, we empirically observed the inductive bias effects as explained in the text.

Finally, in our numerical analysis of fair learning algorithms, we utilized well-known datasets in the fairness literature, including Adult, COMPAS, and CelebA. We note that our numerical analysis only concerned the characteristics of fair learning algorithms and did not attempt to draw any conclusions about the nature of data distribution in these datasets. The COMPAS dataset has been critically analyzed in the machine learning literature [Washington, 2018, Bao et al., 2021], and the connections between the specific dataset and inductive biases of fair learning algorithms will be interesting for future studies.

## ACKNOWLEDGMENTS

The work of Farzan Farnia is partially supported by a grant from the Research Grants Council of the Hong Kong Special Administrative Region, China, Project 14209920, and is partially supported by a CUHK Direct Research Grant with CUHK Project No. 4055164. The work of Amin Gohari is supported by the CUHK Direct Research Grant No. 4055193. Finally, the authors would like to thank the anonymous reviewers for their constructive feedback and helpful suggestions.

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

# 8 APPENDIX

## 8.1 PROOFS

### 8.1.1 Proof of Theorem 1

First, we note the following optimal transport-based formulation of the total variation distance between $P_Y$ and $Q_Y$:

$$\text{TV}(P, Q) = \inf_{\substack{M_{\widehat{Y}, Y} : M_{\widehat{Y}} = P \\ M_Y = Q}} \mathbb{E}_M\left[\ell_{0/1}(\widehat{Y}, Y)\right].$$

Therefore, for the objective function in Equation (1), we can write the following:

$$\mathbb{E}_{P_{\mathbf{X}, \mathbf{Y}, \mathbf{S}, \widehat{\mathbf{Y}}}}\left[\ell_{0/1}(\widehat{Y}, Y)\right] \overset{(a)}{=} \mathbb{E}_{P_{\mathbf{S}}}\left[\mathbb{E}_{P_{\widehat{Y}, Y, \mathbf{X}|S}}\left[\ell_{0/1}(\widehat{Y}, Y)|S = s\right]\right]$$

$$= \mathbb{E}_{P_{\mathbf{S}}}\left[\mathbb{E}_{P_{\widehat{Y}, Y|S}}\left[\ell_{0/1}(\widehat{Y}, Y)|S = s\right]\right]$$

$$\overset{(b)}{\geq} \mathbb{E}_{P_{\mathbf{S}}}\left[\text{TV}\left(P_{\widehat{Y}|S=s}, P_{Y|S=s}\right)\right].$$

Here, (a) follows from the tower property of expectation. Also, (b) is a corollary of the optimal transport formulation of the TV-distance. On the other hand, the constraint in (1) states that $\text{DDP}(\widehat{Y}, S) \leq \epsilon$, implying

$$\mathbb{E}_{P_{\mathbf{S}}}\left[\text{TV}\left(P_{\widehat{Y}|S=s}, P_{\widehat{Y}}\right)\right] = \sum_{s \in \mathcal{S}} P_S(s)\text{TV}\left(P_{\widehat{Y}|S=s}, P_{\widehat{Y}}\right)$$

$$\leq \sum_{s \in \mathcal{S}} \text{TV}\left(P_{\widehat{Y}|S=s}, P_{\widehat{Y}}\right)$$

$$= \frac{1}{2}\text{DDP}(\widehat{Y}, S)$$

$$\leq \frac{\epsilon}{2}.$$

Knowing that TV is a metric distance satisfying the triangle inequality, the above equations show that

$$\mathbb{E}_{P_{\mathbf{X}, \mathbf{Y}, \mathbf{S}, \widehat{\mathbf{Y}}}}\left[\ell_{0/1}(\widehat{Y}, Y)\right] \geq \mathbb{E}_{P_{\mathbf{S}}}\left[\text{TV}\left(P_{\widehat{Y}|S=s}, P_{Y|S=s}\right)\right]$$

$$\overset{(c)}{\geq} \mathbb{E}_{P_{\mathbf{S}}}\left[\text{TV}\left(P_{Y|S=s}, P_{\widehat{Y}}\right) - \text{TV}\left(P_{\widehat{Y}|S=s}, P_{\widehat{Y}}\right)\right]$$

$$= \mathbb{E}_{P_{\mathbf{S}}}\left[\text{TV}\left(P_{Y|S=s}, P_{\widehat{Y}}\right)\right] - \mathbb{E}_{P_{\mathbf{S}}}\left[\text{TV}\left(P_{\widehat{Y}|S=s}, P_{\widehat{Y}}\right)\right]$$

$$\geq \mathbb{E}_{P_{\mathbf{S}}}\left[\text{TV}\left(P_{Y|S=s}, P_{\widehat{Y}}\right)\right] - \frac{\epsilon}{2},$$

where (c) follows from the triangle inequality for TV-distance. Considering the above inequality which holds for every feasible distribution $P_{\widehat{Y}|Y,S}$ satisfying the DDP constraint, we focus on the following specific selection of $Q_{\widehat{Y}|\mathbf{X},Y,S}$. Here we suppose $Q_{\widehat{Y}|S=s} = P_{Y|S=s_{\max}}$ for every $s \in \mathcal{S}$. To find the joint distribution $Q^*_{\widehat{Y},Y|S=s}$ we consider the optimal solution to the following TV-based optimal transport problem for every $s \in \mathcal{S}$

$$Q^*_{\widehat{Y}, Y|S=s} := \underset{\substack{M_{\widehat{Y}, Y} : M_{\widehat{Y}} = P_{Y|S=s_{\max}} \\ M_Y = P_{Y|S=s}}}{\text{argmin}} \mathbb{E}_M\left[\ell_{0/1}(\widehat{Y}, Y)\right].$$

Note that given the above selection of $Q^*_{\widehat{Y}, Y|S}$ and $Q^*_{\widehat{Y}|Y,S} = Q^*_{\widehat{Y}, Y|S}/P_{Y|S}$, we can define the joint distribution $Q_{\mathbf{X}, Y, S, \widehat{Y}} := P_{Y,S} \cdot P_{\mathbf{X}|Y,S}Q^*_{\widehat{Y}|Y,S}$ under which $\mathbf{X} \perp \widehat{Y}|Y, S$. Also, under the defined distribution $Q$, $\widehat{Y}$ and $S$ are independent, and we have

$$\mathbb{E}_{Q_{\widehat{Y}, Y, S}}\left[\ell_{0/1}(\widehat{Y}, Y)\right] = \mathbb{E}_{P_{\mathbf{S}}}\left[\text{TV}\left(P_{Y|S=s}, P_{Y|S=s_{\max}}\right)\right].$$

Therefore, since $\mathbf{X} \perp \widehat{Y} | Y, S$ and $Y = h(\mathbf{X}, S)$ is supposed to be a function of $(\mathbf{X}, S)$, we will further have

$$\mathbb{E}_{Q_{\widehat{Y}, \mathbf{X}, S}}\left[\ell_{0/1}(\widehat{Y}, Y)\right] = \mathbb{E}_{P_{\mathbf{S}}}\left[\mathrm{TV}\left(P_{Y|S=s}, P_{Y|S=s_{\max}}\right)\right].$$

Since $Q_{\widehat{Y}|\mathbf{X}, S}$ is a feasible conditional distribution in the optimization problem 1, we will have

$$\mathbb{E}_{P_{\mathbf{S}}}\left[\mathrm{TV}\left(P_{Y|S=s}, P_{\widehat{Y}}\right)\right] - \frac{\epsilon}{2} \le \mathbb{E}_{Q_{\widehat{Y}, \mathbf{X}, S}}\left[\ell_{0/1}(\widehat{Y}, Y)\right] = \mathbb{E}_{P_{\mathbf{S}}}\left[\mathrm{TV}\left(P_{Y|S=s}, P_{Y|S=s_{\max}}\right)\right].$$

Therefore,

$$
\begin{aligned}
\frac{\epsilon}{2} &\ge \mathbb{E}_{P_{\mathbf{S}}}\left[\mathrm{TV}\left(P_{Y|S=s}, P_{\widehat{Y}}\right)\right] - \mathbb{E}_{P_{\mathbf{S}}}\left[\mathrm{TV}\left(P_{Y|S=s}, P_{Y|S=s_{\max}}\right)\right] \\
&= \sum_{s \in \mathcal{S}} P_S(s)\left(\mathrm{TV}\left(P_{Y|S=s}, P_{\widehat{Y}}\right) - \mathrm{TV}\left(P_{Y|S=s}, P_{Y|S=s_{\max}}\right)\right) \\
&= P_S(s_{\max})\left(\mathrm{TV}\left(P_{Y|S=s_{\max}}, P_{\widehat{Y}}\right) - \mathrm{TV}\left(P_{Y|S=s_{\max}}, P_{Y|S=s_{\max}}\right)\right) \\
&\quad + \sum_{s \ne s_{\max}} P_S(s)\left(\mathrm{TV}\left(P_{Y|S=s}, P_{\widehat{Y}}\right) - \mathrm{TV}\left(P_{Y|S=s}, P_{Y|S=s_{\max}}\right)\right) \\
&= (\frac{1}{2} + \delta)\mathrm{TV}\left(P_{Y|S=s_{\max}}, P_{\widehat{Y}}\right) \\
&\quad + \sum_{s \ne s_{\max}} P_S(s)\left(\mathrm{TV}\left(P_{Y|S=s}, P_{\widehat{Y}}\right) - \mathrm{TV}\left(P_{Y|S=s}, P_{Y|S=s_{\max}}\right)\right) \\
&\stackrel{(d)}{\ge} (\frac{1}{2} + \delta)\mathrm{TV}\left(P_{Y|S=s_{\max}}, P_{\widehat{Y}}\right) - \sum_{s \ne s_{\max}} P_S(s)\mathrm{TV}\left(P_{Y|S=s_{\max}}, P_{\widehat{Y}}\right) \\
&\stackrel{(e)}{=} (\frac{1}{2} + \delta)\mathrm{TV}\left(P_{Y|S=s_{\max}}, P_{\widehat{Y}}\right) - (\frac{1}{2} - \delta)\mathrm{TV}\left(P_{Y|S=s_{\max}}, P_{\widehat{Y}}\right) \\
&= 2\delta\mathrm{TV}\left(P_{Y|S=s_{\max}}, P_{\widehat{Y}}\right).
\end{aligned}
$$

In the above, (d) comes from the triangle inequality for TV-distance, and $(e)$ holds because $\sum_{s \ne s_{\max}} P_S(s) = 1 - P_S(s_{\max}) = \frac{1}{2} - \delta$. The above inequality shows that $\mathrm{TV}\left(P_{Y|S=s_{\max}}, P_{\widehat{Y}}\right) \le \frac{\epsilon}{4\delta}$. We combine this inequality with the DDP constraint, which shows for every $s \in \mathcal{S}$

$$
\begin{aligned}
\mathrm{TV}\left(P_{Y|S=s_{\max}}, P_{\widehat{Y}|S=s}\right) &\le \mathrm{TV}\left(P_{Y|S=s_{\max}}, P_{\widehat{Y}}\right) + \mathrm{TV}\left(P_{\widehat{Y}}, P_{\widehat{Y}|S=s}\right) \\
&\stackrel{(f)}{\le} \frac{\epsilon}{4\delta} + \frac{\epsilon}{2} \\
&= \epsilon\left(\frac{1}{2} + \frac{1}{4\delta}\right).
\end{aligned}
$$

In the above, note that $(f)$ holds because $\mathrm{TV}\left(P_{\widehat{Y}}, P_{\widehat{Y}|S=s}\right) \le \frac{1}{2}\mathrm{DDP}(\widehat{Y}; S) \le \frac{\epsilon}{2}$ according to the optimization constraint. Therefore, the proof is complete.

### 8.1.2 Proof of Theorem 2

We first review the implications of Pinsker's inequality in the cases of mutual information and $\chi^2$-divergence.

**Lemma 1** (Pinsker's inequality for mutual information). *For every pair of random variables $Y, S$, we have*

$$I(Y; S) \ge 2\log(e)\mathbb{E}_S\left[\mathrm{TV}\left(P_{Y|S=s}, P_Y\right)\right]^2.$$

*Proof.* Note that Pinsker's inequality implies that for every outcome $s \in \mathcal{S}$, we have

$$2\log(e)\,\mathrm{TV}\left(P_{Y|S=s}, P_Y\right)^2 \le D_{KL}\left(P_{Y|S=s}, P_Y\right)$$

Since $g(t) = 2\log(e)\,t^2$ is a convex function, Jensen's inequality implies that

$$
\begin{aligned}
2\log(e)\mathbb{E}_S\Big[\text{TV}\big(P_{Y|S=s}, P_Y\big)\Big]^2 &\leq \mathbb{E}_S\Big[2\log(e)\text{TV}^2\big(P_{Y|S=s}, P_Y\big)\Big] \\
&\leq \mathbb{E}_S\Big[D_{KL}\big(P_{Y|S=s}, P_Y\big)\Big] \\
&= I(Y; S).
\end{aligned}
$$

Therefore, the proof is complete. □

**Lemma 2** (Pinsker's inequality for $\chi^2$-divergence-based $f$-mutual-information)**.** *For every pair of random variables $Y, S$, we have the following for function $h(t) = t^2$ where $|t| \leq 1$ and $h(t) = 2t - 1$ where $t \geq 1$.*

$$
\chi^2\big(P_{Y,S}, P_Y \times P_S\big) \geq h\Big(2\mathbb{E}_S\Big[\text{TV}\big(P_{Y|S=s}, P_Y\big)\Big]\Big)
$$

*Proof.* Note that Pinsker's inequality for the $\chi^2$-divergence [Gilardoni, 2006] implies that for every outcome $s \in \mathcal{S}$, we have

$$
h\Big(2\text{TV}\big(P_{Y|S=s}, P_Y\big)\Big) \leq D_{KL}\big(P_{Y|S=s}, P_Y\big)
$$

Since $h$ is a convex function, Jensen's inequality implies that

$$
\begin{aligned}
h\Big(2\mathbb{E}_S\Big[\text{TV}\big(P_{Y|S=s}, P_Y\big)\Big]\Big) &\leq \mathbb{E}_S\Big[h\Big(2\text{TV}\big(P_{Y|S=s}, P_Y\big)\Big)\Big] \\
&\leq \mathbb{E}_S\Big[\chi^2\big(P_{Y|S=s}, P_Y\big)\Big] \\
&= \chi^2\big(P_{Y,S}, P_Y \times P_S\big).
\end{aligned}
$$

Hence, the proof is complete. □

**Lemma 3.** *For every pair of random variables $Y, S$, we have the following for function $h(t) = t^2$ where $|t| \leq 1$ and $h(t) = 2t - 1$ where $t \geq 1$, and constant $r = \min\{|\mathcal{S}|, |\mathcal{Y}|\} - 1$:*

$$
r\rho_m(Y, S) \geq h\Big(2\mathbb{E}_S\Big[\text{TV}\big(P_{Y|S=s}, P_Y\big)\Big]\Big)
$$

*Proof.* The proof follows directly from Lemma 2, noting the following relationship between the maximal correlation $\rho_m(Y, S)$ and the Pearson $\chi^2$-divergence-based $f$-mutual information [Asoodeh et al., 2015]:

$$
r\rho_m(Y, S) \geq \chi^2\big(P_{Y,S}, P_Y \times P_S\big).
$$

□

**Proof for the mutual information case.** Given the mutual information constraint $I(\widehat{Y}, S) \leq \epsilon$ in (1), we can apply Lemma 1 which shows

$$
2\log(e)\mathbb{E}_S\Big[\text{TV}\big(P_{Y|S=s}, P_Y\big)\Big]^2 \leq \epsilon
$$
$$
\Rightarrow \quad \mathbb{E}_S\Big[\text{TV}\big(P_{Y|S=s}, P_Y\big)\Big] \leq \sqrt{\frac{\epsilon}{2\log(e)}}.
$$

Note that we can follow the same proof of Theorem 1, which holds if we change $\text{DDP}(\widehat{Y}, S)$ to $\rho_{TV}(Y, S) := \mathbb{E}_S\Big[\text{TV}\big(P_{\widehat{Y}|S=s}, P_{\widehat{Y}}\big)\Big]$, to prove the following statement:

$$
\mathbb{E}_S\Big[\text{TV}\big(P_{Y|S=s_{\max}}, P_{\widehat{Y}|S=s}\big)\Big] \leq \sqrt{\frac{\epsilon}{2\log(e)}}\Big(1 + \frac{1}{2\delta}\Big) = \sqrt{\frac{2\epsilon}{\log(e)}}\Big(\frac{1}{2} + \frac{1}{4\delta}\Big).
$$

**Proof for the ERMI case $\rho_E$.** Given the constraint $\rho_E(\widehat{Y}, S) \leq \epsilon$ in (1), we can apply Lemma 2 that shows

$$
h\Big(2\mathbb{E}_S\Big[\text{TV}\big(P_{Y|S=s}, P_Y\big)\Big]\Big) \leq \epsilon
$$

$$\Rightarrow \quad 2\mathbb{E}_S\Big[\mathrm{TV}\big(P_{Y|S=s}, P_Y\big)\Big] \leq \max\{\epsilon, \sqrt{\epsilon}\}.$$

In the above, we use the fact that the inverse function of $h(t)$ over $t \geq 0$ satisfies $h^{-1}(t) \leq \max\{t, \sqrt{t}\}$ which is a strictly increasing function. As a result, we can follow the same proof of Theorem 1, which remains valid if we change $\mathrm{DDP}(\widehat{Y}, S)$ to $\rho_{TV}(\widehat{Y}, S) := \mathbb{E}_S\big[\mathrm{TV}\big(P_{\widehat{Y}|S=s}, P_{\widehat{Y}}\big)\big]$, to show the following:

$$\mathbb{E}_S\Big[\mathrm{TV}\big(P_{Y|S=s_{\max}}, P_{\widehat{Y}|S=s}\big)\Big] \leq \max\{\epsilon, \sqrt{\epsilon}\}\Big(\frac{1}{2} + \frac{1}{4\delta}\Big).$$

**Proof for the maximal correlation case $\rho_m$.** Assuming the constraint $\rho_m(\widehat{Y}, S) \leq \epsilon$ in (1), we can apply Lemma 3 showing that

$$h\Big(2\mathbb{E}_S\Big[\mathrm{TV}\big(P_{Y|S=s}, P_Y\big)\Big]\Big) \leq r\epsilon$$
$$\Rightarrow \quad 2\mathbb{E}_S\Big[\mathrm{TV}\big(P_{Y|S=s}, P_Y\big)\Big] \leq \max\{r\epsilon, \sqrt{r\epsilon}\}.$$

As a result, we use the same proof of Theorem 1, that remains valid if we change $\mathrm{DDP}(\widehat{Y}, S)$ to $\rho_{TV}(\widehat{Y}, S) := \mathbb{E}_S\big[\mathrm{TV}\big(P_{\widehat{Y}|S=s}, P_{\widehat{Y}}\big)\big]$, to show

$$\mathbb{E}_S\Big[\mathrm{TV}\big(P_{Y|S=s_{\max}}, P_{\widehat{Y}|S=s}\big)\Big] \leq \max\{r\epsilon, \sqrt{r\epsilon}\}\Big(\frac{1}{2} + \frac{1}{4\delta}\Big).$$

The proof is therefore complete.

### 8.1.3 Proof of Theorem 3

First, we note that under the assumption in Remark 1, there exists a function $\phi : \mathcal{X} \times \mathcal{Y} \to \mathbb{R}$ for which $P_{\mathbf{X}, Y, S}$ satisfies the following equation on the ratio $P_{\mathbf{X}|Y,S}/P_{\mathbf{X}|S}$:

$$\forall \mathbf{x} \in \mathcal{X}, y \in \mathcal{Y}, s \in \mathcal{S}: \quad \frac{P(\mathbf{x} \mid y, s)}{P(\mathbf{x} \mid s)} = \phi(\mathbf{x}, y).$$

The above holds, since given the assumption in Remark 1 we can decompose $\mathbf{X} = [X_0 = g(S), \widetilde{X}]$ such that $\widetilde{X} \perp S$ and $\widetilde{X} \perp S | Y$, implying that

$$\frac{P(\mathbf{x} \mid y, s)}{P(\mathbf{x} \mid s)} = \frac{P([x_0, \widetilde{\mathbf{x}}] \mid y, s)}{P([x_0, \widetilde{\mathbf{x}}] \mid s)}$$
$$= \frac{P(x_0 \mid y, s) P(\widetilde{\mathbf{x}} \mid y, s)}{P(x_0 \mid s) P(\widetilde{\mathbf{x}} \mid s)}$$
$$= \frac{P(x_0 \mid s) P(\widetilde{\mathbf{x}} \mid y)}{P(x_0 \mid s) P(\widetilde{\mathbf{x}})}$$
$$= \frac{P(\widetilde{\mathbf{x}} \mid y)}{P(\widetilde{\mathbf{x}})}.$$

To prove Theorem 3, we can follow the initial steps of Theorem 1's proof, which did not use the assumption $Y = h(\mathbf{X}, S)$, to show the following holds for every feasible distribution $Q_{\widehat{Y}|\mathbf{X},S}$ satisfying the constraint in (2)

$$\mathbb{E}_{P_{\mathbf{X},S}}\Big[\ell_{TV}\big(Q_{\widehat{Y}|\mathbf{X}=\mathbf{x},S=s}, P_{Y|\mathbf{X}=\mathbf{x},S=s}\big)\Big] \geq \mathbb{E}_{P_S}\Big[\mathrm{TV}\big(P_{Y|S=s}, P_{\widehat{Y}}\big)\Big] - \frac{\epsilon}{2}.$$

Next, we consider the following conditional distribution $\widetilde{Q}_{\widehat{Y}|\mathbf{X},S}(y, \mathbf{x}, s) = P_{Y|S=s_{\max}}(y|s_{\max})\phi(\mathbf{x}, \hat{y})$. We note that $\widetilde{Q}_{\widehat{Y}|\mathbf{X},S}$ is a valid conditional distribution under joint distribution $P_{\mathbf{X},S}$ because for every $s \in \mathcal{S}$, $\hat{y} \in \mathcal{Y}$:

$$\sum_{\mathbf{x} \in \mathcal{X}} P_{\mathbf{X}|S}(\mathbf{x}|S = s)\widetilde{Q}_{\widehat{Y}|\mathbf{X},S}(\hat{y}|\mathbf{x}, s) = \sum_{\mathbf{x} \in \mathcal{X}} P_{\mathbf{X}|S}(\mathbf{x}|S = s)P_{Y|S=s_{\max}}(\hat{y}|s_{\max})\phi(\mathbf{x}, \hat{y})$$

$$= P_{Y|S=s_{\max}}(\hat{y}|s_{\max})\sum_{\mathbf{x}\in\mathcal{X}} P_{\mathbf{X}|S}(\mathbf{x}|S=s)\phi(\mathbf{x},\hat{y})$$

$$= P_{Y|S=s_{\max}}(\hat{y}|s_{\max})\sum_{\mathbf{x}\in\mathcal{X}} P_{\mathbf{X}|Y,S}(\mathbf{x}|Y=\hat{y},S=s)$$

$$= P_{Y|S=s_{\max}}(\hat{y}|s_{\max}),$$

which is a valid conditional distribution, implying $\widehat{Y}$ and $S$ are independent under the valid joint distribution $\widetilde{Q}_{\widehat{Y}|\mathbf{X},S}\times P_{\mathbf{X},S}$. Therefore $\widetilde{Q}_{\widehat{Y}|\mathbf{X},S}$ is a feasible conditional distribution in optimization problem (2), implying that under the optimal solution $Q^*_{\widehat{Y}|\mathbf{X},Z}$ we will have

$$\mathbb{E}_{P_{\mathbf{S}}}\Big[\mathrm{TV}\Big(P_{Y|S=s},P_{\widehat{Y}}\Big)\Big] - \frac{\epsilon}{2} \le \mathbb{E}_{P_{\mathbf{X},S}}\Big[\ell_{TV}\big(\widetilde{Q}_{\widehat{Y}|\mathbf{X}=\mathbf{x},S=s},P_{Y|\mathbf{X}=\mathbf{x},S=s}\big)\Big]$$

$$= \mathrm{TV}\big(\widetilde{Q}_{\widehat{Y}|\mathbf{X},S}\times P_{\mathbf{X},S},P_{Y,\mathbf{X},S}\big)$$

$$= \mathrm{TV}\big(P_{Y|S=s_{\max}}\times\phi(\mathbf{X},Y)P_{\mathbf{X},S},P_{Y|S}\times\phi(\mathbf{X},Y)P_{\mathbf{X},S}\big)$$

$$= \mathrm{TV}\big(P_{Y|S=s_{\max}}\times P_S P_{\mathbf{X}|Y,S},P_{Y|S}\times P_S P_{\mathbf{X}|Y,S}\big)$$

$$= \mathrm{TV}\big(P_{Y|S=s_{\max}}\times P_S,P_{Y|S}\times P_S\big)$$

$$= \mathbb{E}_S\Big[\mathrm{TV}\big(P_{Y|S=s_{\max}},P_{Y|S}\big)\Big]$$

As a result, we have the following inequality for the optimal solution $Q^*_{\widehat{Y}|\mathbf{X},S}$ and the constructed $\widetilde{Q}_{\widehat{Y}|\mathbf{X},S}$ resulting in an independent $\widehat{Y}$ of $S$, with the marginal distribution $\widetilde{Q}_{\widehat{Y}} = P_{Y|S=s_{\max}}$:

$$\mathbb{E}_{P_{\mathbf{S}}}\Big[\mathrm{TV}\Big(P_{Y|S=s},P_{\widehat{Y}}\Big)\Big] - \mathbb{E}_S\Big[\mathrm{TV}\big(P_{Y|S=s_{\max}},P_{Y|S}\big)\Big] \le \frac{\epsilon}{2}.$$

Therefore, we can follow the proof of Theorems 1,2 which shows the above inequality leads to the bounds claimed in the theorems.

### 8.1.4 Proof of Theorem 4

We define the function $\phi_s(\mathbf{x},y) := \frac{p(\mathbf{x}|y,s)}{p(\mathbf{x}|s)}$ for the true distribution $P_{\mathbf{X},Y,S}$. Then, in particular,

$$\phi_{s_{\max}}(\mathbf{x},y) = \frac{p(\mathbf{x}|y,s_{\max})}{p(\mathbf{x}|s_{\max})}$$

Note that we can follow the initial steps of the proof of Theorem 1 which does not use the assumption $Y = h(\mathbf{X},S)$, to show the following holds for every $P_{\widehat{Y},\mathbf{X},S} = Q_{\widehat{Y}|\mathbf{X},S}\cdot P_{\mathbf{X},S}$ corresponding to a feasible distribution $Q_{\widehat{Y}|\mathbf{X},S}$ satisfying the constraint in (2)

$$\mathbb{E}_{P_{\mathbf{X},S}}\Big[\ell_{TV}\big(Q_{\widehat{Y}|\mathbf{X}=\mathbf{x},S=s},P_{Y|\mathbf{X}=\mathbf{x},S=s}\big)\Big] \ge \mathbb{E}_{P_{\mathbf{S}}}\Big[\mathrm{TV}\Big(P_{Y|S=s},P_{\widehat{Y}}\Big)\Big] - \mathbb{E}_{P_{\mathbf{S}}}\Big[\mathrm{TV}\Big(P_{\widehat{Y}|S=s},P_{\widehat{Y}}\Big)\Big]$$

$$\ge \mathbb{E}_{P_{\mathbf{S}}}\Big[\mathrm{TV}\Big(P_{Y|S=s},P_{\widehat{Y}}\Big)\Big] - \epsilon.$$

Next, we consider the following conditional distribution

$$\widetilde{Q}_{\widehat{Y}|\mathbf{X},S}(y|\mathbf{x},s) = P_{Y|\mathbf{X},S}(y|\mathbf{x},s_{\max}) = P_{Y|S=s_{\max}}(y|s_{\max})\phi_{s_{\max}}(\mathbf{x},y).$$

Clearly, $\widetilde{Q}_{\widehat{Y}|\mathbf{X},S}$ is a valid conditional distribution. Considering the resulting joint distribution $\widetilde{Q}_{\widehat{Y},\mathbf{X},S} := P_{\mathbf{X},S}\widetilde{Q}_{\widehat{Y}|\mathbf{X},S}$, for every $s\in\mathcal{S}, \hat{y}\in\mathcal{Y}$:

$$\widetilde{Q}_{\widehat{Y}|S}(\hat{y}|s) = \sum_{\mathbf{x}\in\mathcal{X}} P_{\mathbf{X}|S}(\mathbf{x}|S=s)\widetilde{Q}_{\widehat{Y}|\mathbf{X},S}(\hat{y}|\mathbf{x},s)$$

$$= \sum_{\mathbf{x} \in \mathcal{X}} P_{\mathbf{X}|S}(\mathbf{x}|S=s) P_{Y|S=s_{\max}}(\hat{y}|s_{\max}) \phi_{s_{\max}}(\mathbf{x}, \hat{y})$$

$$= P_{Y|S=s_{\max}}(\hat{y}|s_{\max}) \sum_{\mathbf{x} \in \mathcal{X}} P_{\mathbf{X}|S}(\mathbf{x}|S=s) \phi_{s_{\max}}(\mathbf{x}, \hat{y}).$$

According to the triangle inequality for the TV-distance, we have

$$\mathbb{E}_{s \sim P_S}\Big[\mathrm{TV}\big(\widetilde{Q}_{\widehat{Y}|S=s}, \widetilde{Q}_Y\big)\Big] \le \mathbb{E}_{s \sim P_S}\Big[\mathrm{TV}\big(\widetilde{Q}_{\widehat{Y}|S=s}, P_{Y|S=s_{\max}}\big)\Big] + TV\big(P_{Y|S=s_{\max}}, \widetilde{Q}_Y\big)$$

Thus, to show that $\widetilde{Q}_{\widehat{Y}|\mathbf{X},S}$ is a feasible conditional distribution in optimization problem (2) with the TV-based measure $\rho_{TV}$, it suffices to show that

$$\mathbb{E}_{s \sim P_S}\Big[\mathrm{TV}\big(\widetilde{Q}_{\widehat{Y}|S=s}, P_{Y|S=s_{\max}}\big)\Big] \le \frac{\epsilon}{2} \qquad \text{and} \qquad TV\big(P_{Y|S=s_{\max}}, \widetilde{Q}_Y\big) \le \frac{\epsilon}{2}.$$

To show the former, we can write the following inequalities:

$$\left| \widetilde{Q}_{\widehat{Y}|S}(\hat{y}|s) - P_{Y|S}(\hat{y}|s_{\max}) \right| = P_{Y|S=s_{\max}}(\hat{y}|s_{\max}) \left| 1 - \sum_{\mathbf{x} \in \mathcal{X}} P_{\mathbf{X}|S}(\mathbf{x}|S=s) \phi_{s_{\max}}(\mathbf{x}, \hat{y}) \right|$$

$$= P_{Y|S=s_{\max}}(\hat{y}|s_{\max}) \left| \sum_{\mathbf{x} \in \mathcal{X}} P_{\mathbf{X}|S}(\mathbf{x}|S=s) \big( \phi_{s_{\max}}(\mathbf{x}, \hat{y}) - \phi_s(\mathbf{x}, \hat{y}) \big) \right|$$

$$\le \left| \sum_{\mathbf{x} \in \mathcal{X}} P_{Y|S=s_{\max}}(\hat{y}|s_{\max}) P_{\mathbf{X}|S}(\mathbf{x}|S=s) \big( \phi_U(\mathbf{x}, \hat{y}) - \phi_L(\mathbf{x}, \hat{y}) \big) \right|$$

$$= \sum_{\mathbf{x} \in \mathcal{X}} P_{Y|S=s_{\max}}(\hat{y}|s_{\max}) P_{\mathbf{X}|S}(\mathbf{x}|S=s) \Delta(\mathbf{x}, \hat{y}).$$

As a result,

$$\mathbb{E}_{s \sim P_S}\Big[\mathrm{TV}\big(\widetilde{Q}_{\widehat{Y}|S=s}, P_{Y|S=s_{\max}}\big)\Big] \le \sum_s P_S(s) \sum_{\mathbf{x} \in \mathcal{X}, \hat{y} \in \mathcal{Y}} P_{Y|S=s_{\max}}(\hat{y}|s_{\max}) P_{\mathbf{X}|S}(\mathbf{x}|S=s) \Delta(\mathbf{x}, \hat{y})$$

$$= \sum_{\mathbf{x} \in \mathcal{X}, \hat{y} \in \mathcal{Y}} P_{Y|S=s_{\max}}(\hat{y}|s_{\max}) P_{\mathbf{X}}(\mathbf{x}) \Delta(\mathbf{x}, \hat{y})$$

$$= \mathbb{E}_{Y \sim P_{Y|S=s_{\max}}, \mathbf{X} \sim P_{\mathbf{X}}}\Big[\Delta(\mathbf{X}, Y)\Big]$$

$$\le \frac{\epsilon}{2},$$

where the last line follows from the theorem's assumption. Next, we have

$$TV\big(P_{Y|S=s_{\max}}, \widetilde{Q}_Y\big) = \sum_{\hat{y}} P_{Y|S=s_{\max}}(\hat{y}|s_{\max}) \left| \sum_s \sum_{\mathbf{x} \in \mathcal{X}} P_S(s) P_{\mathbf{X}|S}(\mathbf{x}|S=s) \phi_{s_{\max}}(\mathbf{x}, \hat{y}) - 1 \right|$$

$$= \sum_{\hat{y}} P_{Y|S=s_{\max}}(\hat{y}|s_{\max}) \left| \sum_s \sum_{\mathbf{x} \in \mathcal{X}} P_S(s) P_{\mathbf{X}|S}(\mathbf{x}|S=s) \big( \phi_{s_{\max}}(\mathbf{x}, \hat{y}) - \phi_s(\mathbf{x}, \hat{y}) \big) \right|$$

$$\le \sum_{\hat{y}} P_{Y|S=s_{\max}}(\hat{y}|s_{\max}) \left| \sum_s \sum_{\mathbf{x} \in \mathcal{X}} P_S(s) P_{\mathbf{X}|S}(\mathbf{x}|S=s) \Delta(\mathbf{x}, \hat{y}) \right|$$

$$= \mathbb{E}_{Y \sim P_{Y|S=s_{\max}}, \mathbf{X} \sim P_{\mathbf{X}}}\Big[\Delta(\mathbf{X}, Y)\Big]$$

$$\le \frac{\epsilon}{2}.$$

Therefore, $\widetilde{Q}_{\widehat{Y}|\mathbf{X},S}$ is a feasible conditional distribution in optimization problem (2) with a DDP measure $\rho_{TV}$. This fact implies that under the optimal solution $Q^*_{Y|\mathbf{X},Z}$ we will have

$$\mathbb{E}_{P_\mathbf{S}}\Big[\mathrm{TV}\big(P_{Y|S=s}, P_{\widehat{Y}}\big)\Big] - \epsilon \le \mathbb{E}_{P_{\mathbf{X},S}}\Big[\ell_{TV}\big(\widetilde{Q}_{\widehat{Y}|\mathbf{X}=\mathbf{x},S=s}, P_{Y|\mathbf{X}=\mathbf{x},S=s}\big)\Big]$$

$$\begin{aligned}
&= \mathrm{TV}\big(\widetilde{Q}_{\widehat{Y}|\mathbf{X},S} \times P_{\mathbf{X},S}, P_{Y,\mathbf{X},S}\big)\\
&= \mathrm{TV}\big(P_{Y|S=s_{\max}} \times \phi_{s_{\max}}(\mathbf{X},Y)P_{\mathbf{X},S}, P_{Y|S} \times \phi_S(\mathbf{X},Y)P_{\mathbf{X},S}\big)\\
&\leq \mathrm{TV}\big(P_{Y|S=s_{\max}} \times \phi_{s_{\max}}(\mathbf{X},Y)P_{\mathbf{X},S}, P_{Y|S=s_{\max}} \times \phi_S(\mathbf{X},Y)P_{\mathbf{X},S}\big)\\
&\quad + \mathrm{TV}\big(P_{Y|S=s_{\max}} \times \phi_S(\mathbf{X},Y)P_{\mathbf{X},S}, P_{Y|S} \times \phi_S(\mathbf{X},Y)P_{\mathbf{X},S}\big)\\
&\leq \mathbb{E}_{P_{\mathbf{X}}P_{Y|S=s_{\max}}}\Big[\phi_U(\mathbf{X},Y) - \phi_L(\mathbf{X},Y)\Big]\\
&\quad + \mathrm{TV}\big(P_{Y|S=s_{\max}} \times P_S P_{\mathbf{X}|Y,S}, P_{Y|S} \times P_S P_{\mathbf{X}|Y,S}\big)\\
&= \mathbb{E}_{P_{\mathbf{X}}P_{Y|S=s_{\max}}}\Big[\phi_U(\mathbf{X},Y) - \phi_L(\mathbf{X},Y)\Big] + \mathbb{E}_S\Big[\mathrm{TV}\big(P_{Y|S=s_{\max}}, P_{Y|S}\big)\Big]
\end{aligned}$$

As a result, we have the following inequality for the optimal solution $Q^*_{\widehat{Y}|\mathbf{X},S}$ and the constructed $\widetilde{Q}_{\widehat{Y}|\mathbf{X},S}$ resulting in an independent $\widehat{Y}$ of $S$, with the marginal distribution $\mathrm{TV}(\widetilde{Q}_{\widehat{Y}}, P_{Y|S=s_{\max}}) \leq \mathbb{E}_{P_{\mathbf{X}}P_{Y|S=s_{\max}}}[\Delta(\mathbf{X},Y)]$:

$$\begin{aligned}
&\mathbb{E}_{P_{\mathbf{S}}}\Big[\mathrm{TV}\big(P_{Y|S=s}, \widetilde{Q}_{\widehat{Y}}\big)\Big] - \mathbb{E}_{P_{\mathbf{S}}}\Big[\mathrm{TV}\big(P_{Y|S=s_{\max}}, P_{Y|S}\big)\Big] \leq \epsilon + \mathbb{E}_{P_{\mathbf{X}}P_{Y|S=s_{\max}}}\Big[\Delta(\mathbf{X},Y)\Big]\\
\implies &\mathbb{E}_{P_{\mathbf{S}}}\Big[\mathrm{TV}\big(P_{Y|S=s}, P_{Y|S=s_{\max}}\big)\Big] - \mathbb{E}_{P_{\mathbf{S}}}\Big[\mathrm{TV}\big(P_{Y|S=s_{\max}}, P_{Y|S}\big)\Big] \leq \epsilon + 2\,\mathbb{E}_{P_{\mathbf{X}}P_{Y|S=s_{\max}}}\Big[\Delta(\mathbf{X},Y)\Big]
\end{aligned}$$

Therefore, we can repeat the final step of Theorem 1 which shows the above inequality results in the following bound claimed in the theorem:

$$\begin{aligned}
\mathbb{E}_{s \sim P_S}\Big[\mathrm{TV}\big(P_{\widehat{Y}|S=s}, P_{Y|S=s_{\max}}\big)\Big] &\leq \big(1 + \tfrac{1}{2\delta}\big)\Big(\epsilon + 2\mathbb{E}_{P_X \cdot P_{Y|S=s_{\max}}}\big[\Delta(\mathbf{x},y)\big]\Big)\\
&\leq \big(1 + \tfrac{1}{2\delta}\big)2\epsilon
\end{aligned}$$

## 8.2 ADDITIONAL NUMERICAL RESULTS

### 8.2.1 Inductive biases for multi-class sensitive attributes

We perform fair learning experiments on the COMPAS and Adult datasets where instead of a binary $S$, thus we consider a 4-ary sensitive attribute by merging the binary gender and race variables to form a 4-ary sensitive attribute with two different distributions in Figure 7 and Figure 8.

### 8.2.2 Comparison between SA-DRO and imbalanced learning method

We show that one particular advantage of the proposed SA-DRO approach is the method's flexibility in tuning the level of bias reduction, because by varying the DRO coefficient over $[0, \infty)$, the learner can explore the spectrum between the original imbalanced distribution and the fully balanced (uniform) distribution on the sensitive attribute $S$. Please note that the learner will pay the price of addressing the imbalanced distribution by a lower accuracy, and the trade-off between accuracy and bias-reduction could be controlled by varying the coefficient of the DRO regularization term in Figure 9.

### 8.2.3 SA-DRO for distributed image classification

By applying SA-DRO methods on CelebA dataset in federated learning settings, we found that SA-DRO methods achieved a similar accuracy with Client 1's local model while preserving the accuracy for the majority clients and maintaining the same level of DDP, as in Table 3.

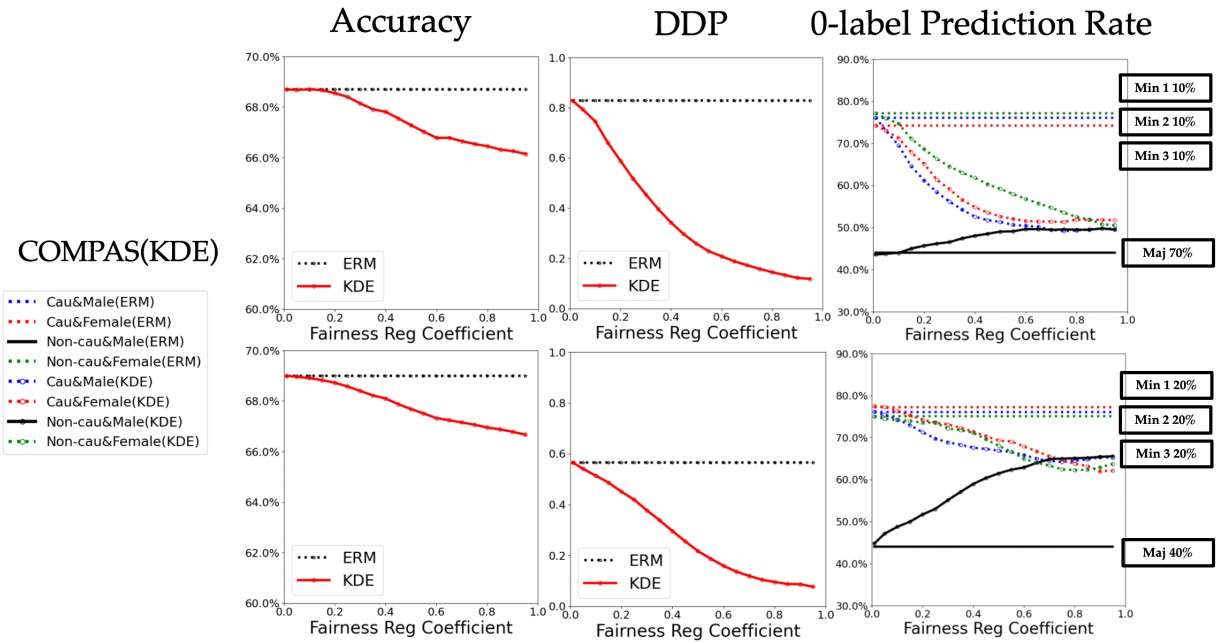

Figure 7: Application of KDE method [Cho et al., 2020b] on COMPAS dataset with multiple sensitive subgroups in two different proportions.

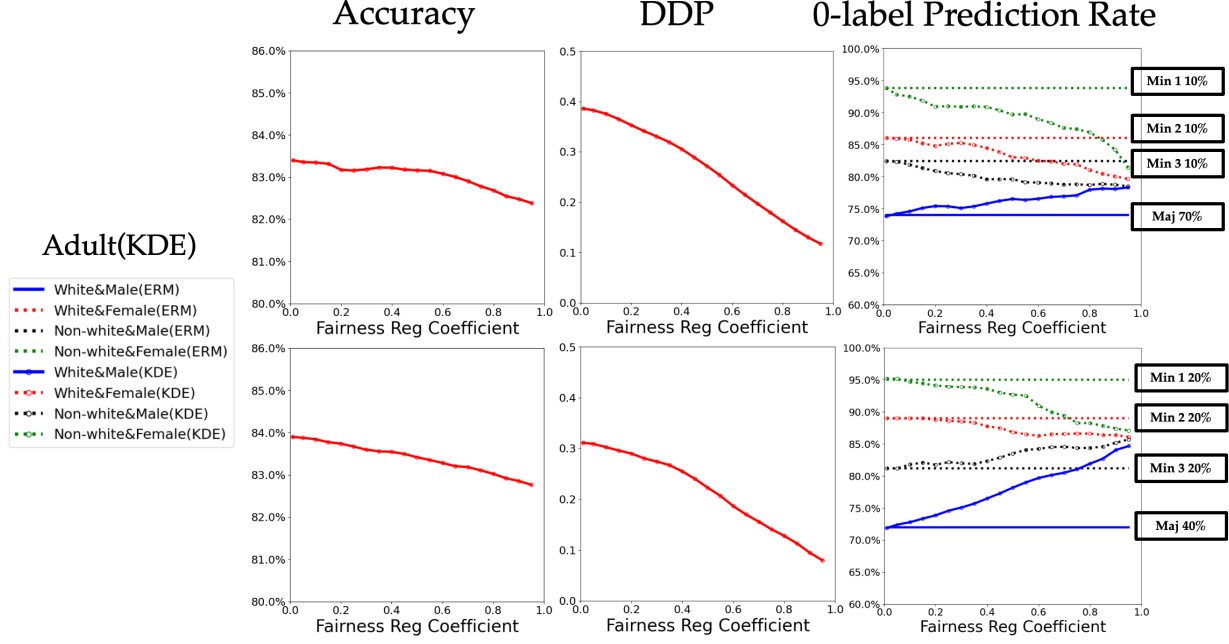

Figure 8: Application of KDE method [Cho et al., 2020b] on Adult dataset with multiple sensitive subgroups in two different proportions.

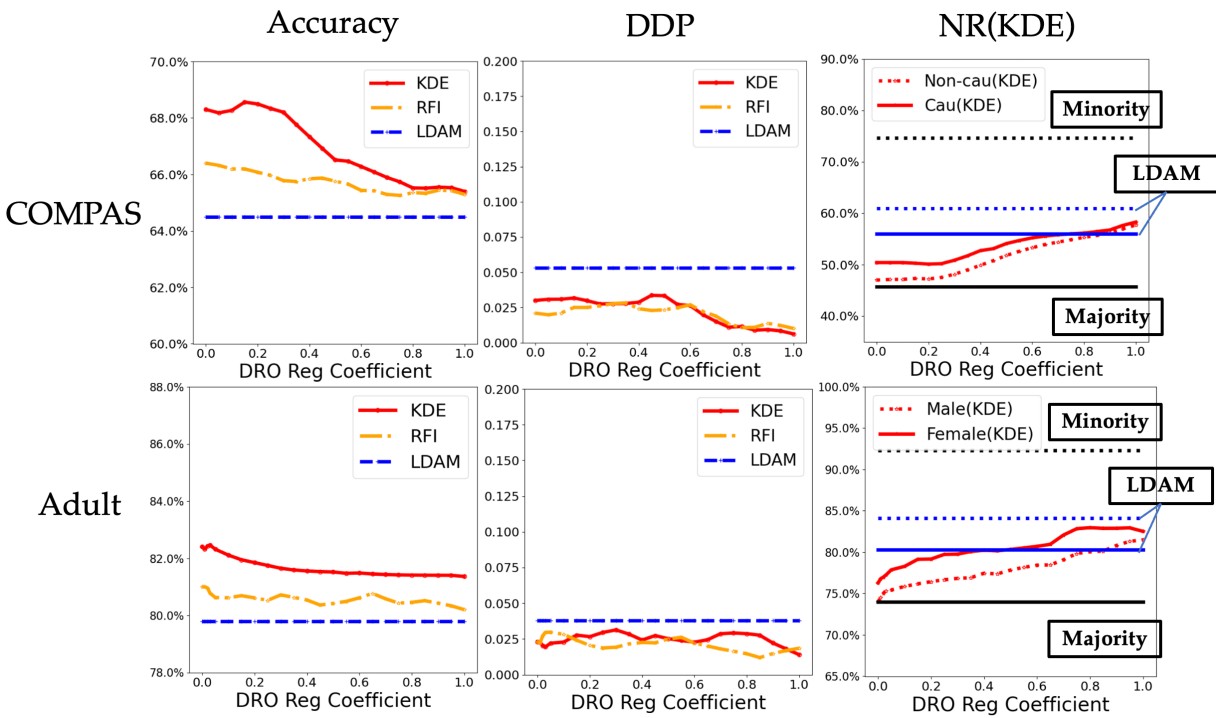

Figure 9: Application of fairness-aware LDAM [Cao et al., 2019] targeting to balance sensitive attributes and compare with SA-DRO.

Table 3: Accuracy and DDP on distributed CelebA dataset

|  | Client 1 (Minority) | | Client 2-5 (Majority) | |
| --- | --- | --- | --- | --- |
|  | Acc($\uparrow$) | DDP($\downarrow$) | Acc($\uparrow$) | DDP($\downarrow$) |
| FedAvg | 94.8% | 0.380 | 94.3% | 0.419 |
| ERM(Local) | 91.8% | 0.374 | 90.3% | 0.396 |
| FedKDE | 65.6% | 0.088 | 88.8% | 0.060 |
| FedFACL | 67.0% | 0.068 | 88.5% | 0.054 |
| **SA-DRO-KDE** | **69.0%** | 0.063 | 88.1% | 0.062 |
| **SA-DRO-FACL** | **68.5%** | 0.057 | 87.7% | 0.069 |
| KDE(Local) | 69.7% | 0.055 | 87.6% | 0.043 |
| FACL(Local) | 69.1% | 0.067 | 87.5% | 0.050 |