# OpenReview forum: "On the Inductive Biases of Demographic Parity-based Fair Learning Algorithms"
_auai.org/UAI/2024/Conference — UAI 2024 poster_

### Official Review · Reviewer_oBdd · 2024-03-03

**Q2-1 Originality-Novelty:** 3
**Q2-2 Correctness-Technical Quality:** 3
**Q2-5 Clarity Of Writing:** 4

**Q10 Ethical Concerns:**

No, but I think the concerns relating to the COMPAS dataset should be acknowleged.

**Q1 Summary And Contributions:**

The authors argue that common approaches to improving demographic parity in classification tasks on datasets with imbalance across the distribution of the sensitive attribute tends to result in a solution biased towards the majority group. This is an important problem, and even on an anecdotal level I see many papers which purport to be 'fair' on the basis of some relatively superficial adaptation of a fairness metric as a regularization technique, without going deeper into whether this leads to other forms of imbalance / inequitable behaviour. This paper addresses a side-effect side-effect of generic approaches to constraining optimisation according to demographic parity.  They provide theoretical proof for this phenomenon, empirical evidence across three datasets, and propose a novel optimization method 'Sensitive Attribute-based Distributionally Robust Optimization (SA-DRO)  to mitigate this issue without significantly impacting accuracy and whilst maintaining low difference in demographic parity (DDP). The method is evaluated and compared against other methods on the COMPAS, Adult, and Celeb-A datasets.

**Q2-3 Extent To Which Claims Are Supported By Evidence:**

3: Good: the main claims are supported by convincing evidence (in the form of adequate experimental evaluation, proofs, (pseudo-)code, references, assumptions).

**Q2-4 Reproducibility:**

3: Good: key resources (e.g. proofs, code, data) are available and key details (e.g. proofs, experimental setup) are sufficiently well-described for competent researchers to confidently reproduce the main results.

**Q3 Main Strengths:**

The paper is very well written, well structured, and easy to follow, and represents a solid balance of theoretical and empirical/experimental content with a proposed solution which is demonstrated to function as expected across an adequate (if not exhaustive) set of evaluations.

One thing I appreciated in particular was the clear link between the theoretical results and the associated empirical demonstrations of the phenomenon, which made a compelling case for why the problem is 'real'.

**Q4 Main Weakness:**

Firstly let me preface this by saying that I have some experience with fairness in machine learning, but it is not my principal area of research, so may not be aware of other work which may already be addressing the issue at hand.

Otherwise, I see little in the way of 'fundamental' weaknesses to address (although I have some smaller points which should be relatively easily addressed in the detailed comments below). I did not thing that the proposed solution was particularly ground-breaking, but I equally do not believe this should be a reason to reject what seems to me to otherwise be a solid contribution. If other reviewers have other perspectives on the novelty of the work relative to recent alternative contributions, I am of course open to new information in this regard.

Following some responses to my detailed comments believe, I believe the current weaknesses can be addressed within the context of a revision.

**Q5 Detailed Comments To The Authors:**

One minor point is that there is no mention of the issues with the COMPASS dataset. I believe this dataset has been heavily criticized so at least some acknowledgement should be made for this criticism, even if the authors see fit to retain the evaluation given its benchmark status.

I have a feeling that Z was used instead of S in a few places (e.g. 2nd column page 3, end of very first paragraph;  also optimization objectives (1) and (2)). I'm not 100% sure if the authors originally used Z and changed it to S but missed some cases, or whether I'm missing a key definition here.

I thought that Figures 2 and 3 were very clear illustrations of the problem being addressed, but was surprised not to see the proposed method evaluated in a similar way until Figure 6 in the Appendix. Firstly, I'm not sure Figure 3 is necessary to keep in the main text, I think this could be in the appendix, and secondly, I think Figure 6 in the Appendix is much more interesting than the current Figure 3, so maybe these could be exchanged?

On a related note, I think Figure 1 does not contribute very much in terms of information, particularly because federated learning ends up a relatively small experimental chunk of the paper, and because it doesn't really communicate the very basics of federated learning than would could be explained in a couple of sentences.  This would free up significant space for more discussion.

And then, following on from 'more discussion', I think it would be good if the authors could provide a motivating example which illustrates the impact of the problem they are solving. Along the lines of some real-world potential issue which could contribute, exacerbate, or perpetuate societal inequity, and which clearly illustrates why existing methods for achieving DP may actually create a new problem along a similar dimension to the one which they actually aim to help.

In Table 2, the DP is not particularly low compared with FedKDE, FedFACL, KDE(local) and FACL(local), I wonder if the authors could comment on this and add something brief to the paper, accordingly.

Finally, the introduction seemed conspicuous in its lack of citations/references. When reading it I felt that there were numerous instances where a reference would be appropriate but there wasn't one. For example, when talking about how biased ML is a problem, I imagine there are many references which document this problem well which should be referenced.

**Q9 Complying With Reviewing Instructions:**

Yes

---

> ### Author Rebuttal · Authors · 2024-04-08
>
> We thank Reviewer oBdd for his/her time, detailed feedback, and constructive suggestions on our work. In the following, we respond to the comments and questions in the review.
>
> **1- Concerns with the COMPAS dataset**
>
> **Re:** We would like to clarify that the focus of our work is only on the general properties of demographic parity-based fair learning methods, and our numerical discussion on the COMPAS and other datasets does not aim to analyze any specific trend in those datasets. We used the Adult, COMPAS, and CelebA datasets in our experimental evaluation, since they commonly appear in the fair learning literature. To address the reviewer’s comment, we will include an ethical statement at the end of the paper including the above information and citing a few well-known studies on the criticisms of COMPAS dataset.
>
> **2- Typos in the notation**
>
> **Re:** We thank the reviewer for catching the typo which we will correct in the revision. As pointed out by the reviewer, the mistakenly written variable $Z$ indeed refers to the sensitive attribute $S$. We apologize for any confusion the typo may have caused.
>
> **3-  Positioning of Figures 2,3,6**
>
> **Re:** Following the reviewer’s suggestion, we will move Figure 6 from the Appendix to the main text which will substitute the current Figure 3 that we will postpone to the Appendix.
>
> **4- Including motivating examples in the introduction**
>
> **Re:** A motivating example would be a hiring task where the label $Y$ (showing whether the applicant is qualified or not) correlates with a binary sensitive attribute $S\in \{ a,b \}$, i.e. the training data suggest that applicants from subgroup $S= a$ are qualified more often (on average) than the applicants from $S=b$. Then, if the target population includes significantly more applicants from subgroup $a$ than subgroup $b$, a DP-based fair classifier would be inductively biased to extend the hiring rate for the majority subgroup $a$ to subgroup $b$. Please note that such extreme examples of the biases of fair learning are to some extent known in the community, e.g. the reviewer may refer to the introduction of (Hardt at el, 2016). Our main contribution is to mathematically formulate the inductive biases that could impact a DP-based fair learning algorithm. We will discuss the example in the introduction.
>
> **5- Federated learning example in Figure 1**
>
> **Re:** Following the reviewer’s comment, we will move Figure 1 to Section 7. As the reviewer pointed out, our work does not focus on fairness in federated learning; however, we would like to highlight that the phenomenon illustrated in Figure 1 could be problematic in a general distributed learning scenario where the data may come from various sources with potentially different marginal sensitive attribute distributions.
>
> For example, one can consider a scenario where the departments of a university or company plan to cooperate to make fair hiring decisions and run a DP-based fair learning algorithm on all the departments’ collected applications. If one or a few of the involved departments receive significantly more frequent applications from the minority sensitive attribute (e.g. a particular gender or race), those departments could face a disproportionate loss in the ratio of qualified hired applicants. Such impacts of a naive application of DP-based fair learning algorithms to distributed learning problems could serve as a motivation for our study on the inductive biases of DP-based fair learning algorithms.
>
> **6- Table 2**
>
> **Re:** Since we did not change the fairness regularization hyperparameter $\lambda$ in the experiments of Figure 2, varying the DRO regularization coefficient led to only a little worse DP fairness compared to the original value. We reported the performance when we matched the minority client-based localized accuracy with the client’s locally trained fair model and that is why the DDP fairness degraded a little. To address the reviewer’s comment, we increased the $\lambda$ coefficient to force the DDP fairness violation below 3%, and the updated table is presented below, showing a slight decrease in the accuracy of minority users but less significant than in the non-DRO fair federated training. We will clarify the comparison in the text.
>
> |      Methods      |  Acc(Minority)  |  DDP(Minority)  |  Acc(Majority)  |  DDP(Majority)  |
> |:-----------------:|:-----:|:-----:|:-----:|:-----:|
> | **SA-DRO-FedKDE** | **79.3%** | **0.041** | 89.6% | 0.042 |
> | **SA-DRO-FedFACL**  | **79.0%** | **0.055** | 89.2% | 0.038 |
> | **SA-DRO-FedKDE(New)** | **78.5%** | **0.028** | 89.0% | 0.032 |
> | **SA-DRO-FedFACL(New)**  | **78.3%** | **0.030** | 88.7% | 0.027 |
> |     KDE(Local)    | 79.0% | 0.032 | 88.2% | 0.014 |
> |    FACL(Local)    | 79.1% | 0.025 | 88.6% | 0.017 |
>
> **7- References in the Introduction**
>
> **Re:** To address the reviewer’s comment, we will include more references on DP-based fair learning algorithms and bias reduction methods in the introduction.

---

### Official Review · Reviewer_TSpv · 2024-03-22

**Q2-1 Originality-Novelty:** 3
**Q2-2 Correctness-Technical Quality:** 3
**Q2-5 Clarity Of Writing:** 3

**Q1 Summary And Contributions:**

This paper investigates the challenge of ensuring fairness in supervised learning algorithms, focusing on those that aim to be independent of sensitive attributes. The demographic parity (DP) measure, commonly used to assess fairness, is critiqued for potentially introducing bias when applied to datasets with non-uniform distributions of sensitive attributes. The authors analytically explore how standard DP-based regularization can skew a model toward the majority class of a sensitive attribute in imbalanced datasets. To counter this, they propose a sensitive attribute-based distributionally robust optimization (SA-DRO) method, enhancing robustness to varying distributions of sensitive attributes. This paper's key contributions include this novel SA-DRO method and several numerical experiments demonstrating the method's effectiveness in mitigating biases in both centralized and distributed learning environments, thereby supporting the theoretical findings about biases in DP-based fairness approaches.

**Q2-3 Extent To Which Claims Are Supported By Evidence:**

3: Good: the main claims are supported by convincing evidence (in the form of adequate experimental evaluation, proofs, (pseudo-)code, references, assumptions).

**Q2-4 Reproducibility:**

3: Good: key resources (e.g. proofs, code, data) are available and key details (e.g. proofs, experimental setup) are sufficiently well-described for competent researchers to confidently reproduce the main results.

**Q3 Main Strengths:**

This paper focuses on a specific and common problem in the field of fairness, the demographic parity (DP) measure introduces bias when applied to datasets with non-uniform distributions of sensitive attributes. This paper is well-organized and clearly written.

**Q4 Main Weakness:**

1. Related works can focus on discussing the problem of data imbalance in fair machine learning.
2. In your algorithm, step 3, what does T stand for?
3. What is your intuition for using mutual information? In addition to mutual information, can other methods be used, such as KL divergence? What is the difference in performance between the two?

**Q5 Detailed Comments To The Authors:**

Please check for weaknesses.

**Q9 Complying With Reviewing Instructions:**

Yes

---

> ### Author Rebuttal · Authors · 2024-04-08
>
> We thank Reviewer TSpv for his/her time and constructive feedback and suggestions on our work. In the following, we respond to the comments and questions in the review.
>
> **1- Related works on data imbalance in fair machine learning**
>
> **Re:** We thank the reviewer for pointing this out. In the revision, we will discuss the following references related to data imbalance in fair machine learning. Please let us know if we have missed any other related work.
>
> - Iosifidis, V., & Ntoutsi, E. (2020, October). -Online Fairness-Aware Learning Under Class Imbalance. In International Conference on Discovery Science (pp. 159-174). Cham: Springer International Publishing.
> - Subramanian, S., Rahimi, A., Baldwin, T., Cohn, T., & Frermann, L. (2021). Fairness-aware class imbalanced learning. arXiv preprint arXiv:2109.10444.
> - Deng, Z., Zhang, J., Zhang, L., Ye, T., Coley, Y., Su, W. J., & Zou, J. (2022). Fifa: Making fairness more generalizable in classifiers trained on imbalanced data. arXiv preprint arXiv:2206.02792.
> - Tarzanagh, D. A., Hou, B., Tong, B., Long, Q., & Shen, L. (2023, July). Fairness-aware class imbalanced learning on multiple subgroups. In Uncertainty in Artificial Intelligence (pp. 2123-2133). PMLR.
>
> **2- $T$ in Algorithm 1**
>
> **Re:** $T$ is the total number of iterations of running the gradient-based min-max optimization algorithm to solve the SA-DRO fair learning task. We will clarify the notation in the text.
>
> **3- Mutual information and other KL-divergence-based dependence measures**
>
> **Re:** We note that applying mutual information as a dependence measure in the fair learning problem has been proposed in several related works, and our analysis of mutual information is based on those proposed applications. We understand the reviewer’s idea of defining other KL-divergence-based dependence metrics, and would like to highlight the role of Pinsker’s inequality (used in the proof of Theorem 2) which can be similarly applied to extend Theorem 1 to other KL-divergence-based dependence metrics. We will explain more about such potential generalizations of Theorem 2 to other KL-based dependence measures after Theorem 2.

---

### Official Review · Reviewer_x5j7 · 2024-03-25

**Q2-1 Originality-Novelty:** 2
**Q2-2 Correctness-Technical Quality:** 3
**Q2-5 Clarity Of Writing:** 3

**Q1 Summary And Contributions:**

This paper studies the DP-based regularization methods under non-uniform distribution given the sensitive attribute. Specially, they show the potential classification bias towards sensitive attributes and thus proposed a distributionally robust optimization to improving robustness against the marginal distribution of the sensitive attribute. Empirical results demonstrate the effect of the proposed method over the current state-of-the-art baselines.

**Q2-3 Extent To Which Claims Are Supported By Evidence:**

3: Good: the main claims are supported by convincing evidence (in the form of adequate experimental evaluation, proofs, (pseudo-)code, references, assumptions).

**Q2-4 Reproducibility:**

2: Fair: key resources (e.g. proofs, code, data) are unavailable but key details (e.g. proof sketches, experimental setup) are sufficiently well-described for an expert to confidently reproduce the main results.

**Q3 Main Strengths:**

Generally, the authors focus on the robustness of the current DP-based methods for fair supervised learning under data imbalance or skew.
- The authors provided the analytical analysis on the inductive biases of DP-based fair learning toward the majority sensitive attribute,
- The authors introduced a a distributionally robust optimization method to lower the biases of DP-based fair classification,
- The authors conducted a range of experiments to support their claim and verify the effectiveness of the proposed method in centralized and federated learning scenarios.

**Q4 Main Weakness:**

Despite the effectiveness, I still have some general concerns about the current methods.

- It is common that the data may follow the skewed distribution, which motivates the explorations of imbalanced learning. The authors have not considered the common way (e.g., the robust imbalance loss like LA[1]) for imbalanced learning in the framework of DP-based methods. The inductive biases may be easily addressed when applying some robust imbalanced loss instead of the ERM loss.

- It will be better to conduct an extended experiments on a larger real-world dataset (may be easily skew in distribution).

**Q5 Detailed Comments To The Authors:**

See above weakness.

**Q9 Complying With Reviewing Instructions:**

Yes

---

> ### Author Rebuttal · Authors · 2024-04-08
>
> We thank Reviewer x5j7 for his/her time and constructive feedback and suggestions on our work. In the following, we respond to the comments and questions in the review.
>
> **1- Other existing methods to learn on an imbalanced dataset**
>
> **Re:** We agree with the reviewer that one can utilize existing non-DRO methods for learning under an imbalanced dataset. One particular advantage of the DRO approach is the method’s flexibility in tuning the level of bias reduction, because by varying the DRO coefficient over $[0,\infty)$, the learner can explore the spectrum between the original imbalanced distribution and the fully balanced (uniform) distribution on the sensitive attribute $S$. Please note that the learner will pay the price of addressing the imbalanced distribution by a lower accuracy, and the trade-off between accuracy and bias-reduction could be controlled by varying the coefficient of the DRO regularization term.
>
> Based on the reviewer’s suggestion, we tested the baseline LDAM method and our experimental results indicate that the LDAM method can fully remove the inductive biases at a little higher accuracy price compared to the SA-DRO method, since the SA-DRO seems to have more flexibility in exploring the accuracy-inductive bias trade-off. The numerical results are available at the [anonymous GitHub repository](https://github.com/lllaaaa123/Anonymous-A738), which we will add to the revised draft.
>
> **2- Experiments on large-scale datasets**
>
> **Re:** We would like to refer the reviewer to the submitted Appendix where we present the experimental results on the more large-scale CelebA dataset. The CelebA dataset contains a high-dimensional feature vector of size $3\times 128\times 128$, and we used the large-scale ResNet18 neural net architecture as the classifier in those experiments. To address the reviewer’s comment, we will move the CelebA numerical results from the Appendix to the main text.

---

### Official Review · Reviewer_oGAd · 2024-03-26

**Q2-1 Originality-Novelty:** 3
**Q2-2 Correctness-Technical Quality:** 2
**Q2-5 Clarity Of Writing:** 4

**Q1 Summary And Contributions:**

This paper studied fair classification learning aiming to satisfy demographic parity. Focusing on in-processing approaches, the authors show that under some assumptions about the target label, there is inductive bias to have the classification probability of each sensitive subgroup to be close to the label probability of the majority group. This bias is shown to exist in both the direct regularization-based method as well as in indirect, dependence-measure based approaches. They then propose a sensitive attribute-based distributionally robust optimization (SADRO) method to reduce this bias and have the classification probabilities of each sensitive subgroup to tend towards “meeting in the middle”.

**Q2-3 Extent To Which Claims Are Supported By Evidence:**

3: Good: the main claims are supported by convincing evidence (in the form of adequate experimental evaluation, proofs, (pseudo-)code, references, assumptions).

**Q2-4 Reproducibility:**

3: Good: key resources (e.g. proofs, code, data) are available and key details (e.g. proofs, experimental setup) are sufficiently well-described for competent researchers to confidently reproduce the main results.

**Q3 Main Strengths:**

The paper makes an interesting theoretical contribution characterizing the inductive bias of DP-based fair learning, as well as introducing a new method to address it. The application of distributionally robust optimization to address this was also interesting.

The paper is clearly written and easy to follow.

Empirical evaluation is thorough, demonstrating effectiveness using three benchmark datasets and multiple baseline methods.

**Q4 Main Weakness:**

There may be a minor error in Theorems 1 and 2 (constant factor in the bounds). Please see detailed comments.

The theoretical results are in terms of learning formulation that minimizes TV distance (Eq (2)) instead of metrics like cross entropy or KL divergence which are more commonly used for learning.

One weak point regarding the experiments is that they are set up such that there are only two subgroups with a huge 80/20 imbalance. The results would be more interesting with multiple subgroups and a more realistic distribution.

**Q5 Detailed Comments To The Authors:**

In p.11 of the Appendix 8.1.1, in the equation before the sentence starting “Knowing that TV…”: I think the equation $\sum_s TV(\dots) = DDP(\dots)$ is missing a factor of 1/2 using the definition of TV and DDP in this paper. This would affect the bounds in both Theorem 1 and 2. I hope the authors can clarify if my understanding is incorrect.

For generalizing the theorems to the setting where labels are not a deterministic function of the inputs, Assumption 1 is very strong. If this holds, then there is a set of features $\tilde{\mathbf{X}}$ independent of the sensitive attribute that can be used to train a classifier and achieve demographic parity. Remark 1 does relax this a little bit, but a bit more discussion on the intuition of the assumption and when it holds would be helpful. I would also suggest directly stating Theorem 3 in terms of the more general assumption in Remark 1 rather than Assumption 1.

Does the assumption in Remark 1 (approximately) hold in the datasets used in experiments?

The inductive bias that enforcing DP tends to make the classification rates match that of the majority group is not very surprising as that is the intuition behind DP: the minority groups get the preferable decision with lower probability than the majority, which we want to fix. Can anything be said with multiple sensitive groups where no single group covers more than 50% of the population?

Minor comments/typos:
- p.3: “dependence on Z” -> “dependence on S”
- First paragraph of p.8: unreferred table “in this table”
- Same paragraph: What does the DRO regularization coefficient $\epsilon = 0.9$ refer to? Is it $\delta$ in Eq (3)?
- Figures 2 and 3: I would suggest showing the accuracy and DDP of ERM as baselines similar to the prediction rate plots.

**Q9 Complying With Reviewing Instructions:**

Yes

---

> ### Author Rebuttal · Authors · 2024-04-08
>
> We thank Reviewer oGAd for his/her time, detailed feedback, and constructive suggestions on our work. In the following, we provide our response to the comments and questions in the review.
>
> **1- Missing constant 2 in the equation relating TV distance and DDP**
>
> **Re:** We thank the reviewer for mentioning the missing constant 2 in that equation. As pointed out, the reduction of DDP to the sum of total variation distances requires an additional multiplicative constant 2. We will revise Theorem 1 accordingly.
>
> **2- Theoretical results on the expected TV loss in the learning task**
>
> **Re:** As mentioned, Theorem 3 is shown for Problem (2) considering the total variation distance between the conditional distributions. Since TV distance is the optimal transport cost for 0/1 loss function, Problem (2) gives an extension of Problem (1) formulated with 0/1 loss. We note that our analysis utilizes the metric property of TV, and thus does not apply to non-metric KL-divergence. Extending the analysis to non-metric divergences is an interesting future direction.
>
> **3- The assumption in Remark 1**
>
> **Re:** After the original submission of the paper, we realized that the assumption in Remark 1 can be relaxed by making minor changes to the proof. In fact, a version of Theorem 3 continues to hold under the following relaxation of Remark 1's assumption: Let $\phi_L : \mathcal{X}\times \mathcal{Y} \rightarrow \mathbb{R}$ and $\phi_U : \mathcal{X}\times \mathcal{Y} \rightarrow \mathbb{R}$ be two functions for which the following inequality holds for every $\mathbf{x} , y ,s$:
>
> $$ \phi_L(\mathbf{x},y) \le \frac{p(\mathbf{x} | y,s)}{p(\mathbf{x} | s)} \le \phi_U(\mathbf{x},y)$$
>
> In particular, the above inequalities will hold if we define $\phi_L (\mathbf{x},y) = \min_s \frac{p(\mathbf{x} | y,s)}{p(\mathbf{x} | s)}$ and $\phi_U (\mathbf{x},y) = \max_s \frac{p(\mathbf{x} | y,s)}{p(\mathbf{x} | s)} $.
> Then, under the assumption that $\epsilon \ge 2\mathbb{E}\bigl[ \phi_U(\mathbf{X},Y)-\phi_L(\mathbf{X},Y)\bigr]$ (expectation according to $P_{X}P_{Y\vert S=s_{\max}}$), the proof of Theorem 3 will go through if we replace $\epsilon$ with $\epsilon +2\mathbb{E}[ \phi_U(\mathbf{X},Y)-\phi_L(\mathbf{X},Y)]$ in the theorem’s conclusion. We have put the drafted proof for the extended version
> in the [anonymous GitHub repository](https://github.com/lllaaaa123/Anonymous-A738/tree/42e45c15063c06a81d9a1068da35725490f40f50/proof). The reviewer’s feedback on the extended version is much appreciated, and we will include the extended version in the submitted paper if the reviewers do not raise any concerns about the modification.
>
> **4- "Does the assumption in Remark 1 (approximately) hold in the datasets used in experiments?"**
>
> **Re:** Since the sample size of the COMPAS and Adult datasets is insufficient for estimating the feature vector’s distribution $P_{\mathbf{X}}$, we were unable to verify whether the assumption holds in the experiments. However, we still observed the effects of the inductive bias in the experiments. We will discuss this limitation of our numerical analysis in a section titled “Limitations and Broader Impact” at the end of the revised draft.
>
> **5- "Can anything be said with multiple sensitive groups where no single group covers more than 50% of the population?"**
>
> **Re:** Yes. In the general case, Corollary 1 (assuming full independence between $\hat{Y}$ and $S$) can be extended to show that the conditional distribution $P_{\hat{Y} |S}$ of the decision variable $\hat{Y}$ given sensitive attribute $S$ will be inductively biased toward the *geometric median* (according to the TV-distance) of the underlying conditional distributions  $P_{Y |S=s}$ where $s\sim P_S$. We will add the remark on the geometric median-based extension of the result after Corollary 1.
>
> **6- Typos**
>
> **Re:** We thank the reviewer for catching the typos. We will correct them in the revision.
>
> **7- DRO Regularization coefficient**
>
> **Re:** We would like to clarify that following the commonly-used implementation of DRO, we used a Lagrangian penalty term $-\zeta d(P_S , Q_S)$ in the inner maximization problem to perform DRO. Therefore, by the coefficient of DRO regularization, we mean the Lagrangian multiplier $\zeta$. We will make this point clear in the text and Algorithm 1.
>
> **8- Figures 2,3**
>
> **Re:** We thank the reviewer for the suggestion. We will add the reference ERM bars to the accuracy and DDP plots in Figures 2,3.
>
> **9- Experiments with more than 2 sub-groups**
>
> **Re:** Based on the comment, we have performed fair learning experiments on the COMPAS and Adult datasets where instead of a binary $S$, we consider a 4-ary sensitive attribute by merging the binary gender and race variables to form a 4-ary sensitive attribute. The numerical findings, available at [anonymous GitHub repository](https://github.com/lllaaaa123/Anonymous-A738) looked mostly consistent with our theoretical results. We will include the numerical results in the text.

---

### Meta-Review · Area_Chair_ryX3 · 2024-04-16

This paper provides an in-depth analysis of the inductive bias introduced by using demographic-parity regularization for training a fair classifier. The authors focuses on the in-processing approaches, and revealed an inductive bais of driving the classification probability toward the ones for the majority group. This bias shown to exist in both regularization based methods as well as in other indirect approaches that use dependence measures. The paper then proposes a sensitive attribute-based distributionally robust optimization (SA-DRO) method to mitigate the impact of this bias.

This paper made an interesting and important theoretical contribution towards unpacking the working mechanisms of the fairness constrained solutions. The proposed SA-DRO contributes to the literature a robust solution to improving model fairness when facing imbalance data. Overall, reviewers are positive about the work’s technical quality and the paper is well written and easy to follow.